# Scalar products and norm of Bethe vectors
# for integrable models based on $U_q(\widehat{\mathfrak{gl}}_m)$

**Arthur Hutsalyuk[1,2]★, Andrii Liashyk[3,4,5]♣, Stanislav Z. Pakuliak[1,6]†, Eric Ragoucy[7]‡ and Nikita A. Slavnov[8]◇**

**1** Moscow Institute of Physics and Technology, Dolgoprudny, Moscow reg., Russia
**2** Fachbereich C Physik, Bergische Universität Wuppertal, 42097 Wuppertal, Germany
**3** Bogoliubov Institute for Theoretical Physics, NAS of Ukraine, Kiev, Ukraine
**4** National Research University Higher School of Economics, Faculty of Mathematics, Moscow, Russia
**5** Skolkovo Institute of Science and Technology, Moscow, Russia
**6** Laboratory of Theoretical Physics, JINR, Dubna, Moscow reg., Russi
**7** Laboratoire de Physique Théorique LAPTh, CNRS and USMB, BP 110, 74941 Annecy-le-Vieux Cedex, France
**8** Steklov Mathematical Institute of Russian Academy of Sciences, Moscow, Russia

★ hutsalyuk@gmail.com, ♣ a.liashyk@gmail.com, † stanislav.pakuliak@jinr.ru, ‡ eric.ragoucy@lapth.cnrs.fr, ◇ nslavnov@mi.ras.ru

## Abstract

**We obtain recursion formulas for the Bethe vectors of models with periodic boundary conditions solvable by the nested algebraic Bethe ansatz and based on the quantum affine algebra $U_q(\widehat{\mathfrak{gl}}_m)$. We also present a sum formula for their scalar products. This formula describes the scalar product in terms of a sum over partitions of the Bethe parameters, whose factors are characterized by two highest coefficients. We provide different recursions for these highest coefficients.**

**In addition, we show that when the Bethe vectors are on-shell, their norm takes the form of a Gaudin determinant.**

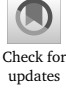

# 1 Introduction

Integrable models have the striking property that their physical data are exactly computable, without the use of any perturbative expansion or asymptotic behavior. For this reason, they have always attracted the attention of researchers. In the twentieth century, quantum integrable models have been the source of many developments originating in the so-called Bethe ansatz, introduced by H. Bethe [1]. In a few words, the Bethe ansatz is an expansion of Hamiltonian eigenvectors over some clever basis (similar to planar waves) using some parameters (the Bethe parameters, which play the role of momenta). Demanding the vectors to be eigenvectors of the Hamiltonian leads to a quantization of the Bethe parameters which takes the form of a system of coupled algebraic equations called the Bethe equations. Knowing the form of the Bethe ansatz and the Bethe equations is in general enough to get a large number of information on the physical data of the system.

In continuity to the Bethe ansatz technics, the Quantum Inverse Scattering Method (QISM), mainly elaborated by the Leningrad/St-Petersburg School [2–5], has been the core of a wide range of progress. These developments were performed in continuity with (or parallel to) the works of C. N. Yang, R. Baxter, M. Gaudin, and many others, see e.g. [6–12].

The Bethe ansatz and QISM have provided a lot of interesting results for the models based on $\mathfrak{gl}_2$ symmetry and its quantum deformations. Among them, we can mention the determinant representations for the norm and the scalar products of Bethe vectors [13, 14]. Focusing on spin chains with periodic boundary conditions, it is worth mentioning the explicit solution of the quantum inverse scattering problem [15–17]. These results were used to study correla-

tion functions of quantum integrable models in the thermodynamic limit via multiple integral representations [18–20] or form factor expansion [21–23].

For higher rank algebras, that is to say for multicomponent systems, $\mathfrak{gl}_m$ spin chains and their quantum deformation, or their $\mathbb{Z}_2$-graded versions, results are scarcer, although the general ground has been settled many years ago [24–29]. Nevertheless, some steps have been done, in particular for models with periodic boundary conditions: an explicit expression for Bethe vectors of models based on $Y(\mathfrak{gl}(m|n))$ and on $U_q(\widehat{\mathfrak{gl}}_m)$ can be found in [30–32] and [33–37]. The calculation of scalar product and form factors have been addressed for some specific algebras. The case of the $Y(\mathfrak{gl}_3)$ algebra has been studied in a series of works presenting some explicit forms of Bethe vectors [38], the calculation of their scalar product [39–43] and the expression of the form factors as determinants [44, 45]. Results for models based on the deformed version $U_q(\widehat{\mathfrak{gl}}_3)$ have been also obtained: explicit forms of Bethe vectors can be found in [46], their scalar products in [47–49] and a determinant expression for scalar products and form factors of diagonal elements was presented in [50]. The supersymmetric counterpart of $Y(\mathfrak{gl}_3)$, the superalgebra $Y(\mathfrak{gl}(2|1))$ has been dealt in [51–54]. Some partial results were also obtained for superalgebras in connection with the Super-Yang–Mills theories [55–57]. However a full understanding of the general approach to compute correlation functions is still lacking. Recently, some general results on the scalar product and the norm of Bethe vectors for $Y(\mathfrak{gl}(m|n))$ models have been obtained in [58, 59], in parallel to the original results described in [13, 39]. The present paper contains similar results for models based on the quantum affine algebra $U_q(\widehat{\mathfrak{gl}}_m)$.

It is known (see e.g. [13, 14, 60]) that most of the results concerning the scalar products of Bethe vectors in the models described by the $Y(\mathfrak{gl}_2)$ and $U_q(\widehat{\mathfrak{gl}}_2)$ algebras can be formulated in a sole universal form. This is because the $R$-matrices in both cases correspond to the six-vertex model. An analogous similarity takes place in the general $Y(\mathfrak{gl}_m)$ and $U_q(\widehat{\mathfrak{gl}}_m)$ cases. In spite of some differences between the $R$-matrices of $Y(\mathfrak{gl}_m)$ and $U_q(\widehat{\mathfrak{gl}}_m)$ based models the general structure for the recursions on Bethe vectors, their scalar products, and the properties of the scalar product highest coefficients, is almost identical. Moreover, most proofs literally mimic each other for both cases. Thus, we do not reproduce the proofs entirely, referring the reader to the works [58, 59] for the details. Instead, we mostly focus on the differences between these two cases.

The plan of the article is as follows. We describe our general framework in the two first sections: section 2 contains the algebraic framework used to handle integrable models, and section 3 gathers some properties of the Bethe vectors of $U_q(\widehat{\mathfrak{gl}}_m)$ based models. Section 4 presents our results, which are of two types. Firstly, we show results obtained for generic Bethe vectors: several recursion formulas for the Bethe vectors (section 4.1); a sum formula for their scalar products (section 4.2); and properties of the scalar product highest coefficients (section 4.3). Secondly, considering on-shell Bethe vectors, we give a determinant form *à la Gaudin* for their norm (section 4.4). The following sections are devoted to the proofs of our results. Section 5 deals with the Bethe vectors constructed within the algebraic Bethe ansatz and presents the proofs for the results given in section 4.1. Section 6 contains the proof of the sum formula, and in section 7 we consider the symmetry properties of the highest coefficients. Appendix A presents the explicit construction of Bethe vectors in a particular simple case. Some of the results obtained in the present paper were already presented in the case of $U_q(\widehat{\mathfrak{gl}}_3)$ in different articles: we make the connection with them in appendix B. A coproduct property for dual Bethe vectors is proven in appendix C.

## 2 Description of the model

### 2.1 The $U_q(\widehat{\mathfrak{gl}}_m)$ based quantum integrable model

Let $R(u, v)$ be a matrix associated with the vector representation of the quantum affine algebra $U_q(\widehat{\mathfrak{gl}}_m)$:

$$
\begin{aligned}
R(u, v) = {} & f(u, v) \sum_{1 \le i \le m} E_{ii} \otimes E_{ii} + \sum_{1 \le i < j \le m} \left( E_{ii} \otimes E_{jj} + E_{jj} \otimes E_{ii} \right) \\
& + \sum_{1 \le i < j \le m} g(u, v) \left( u E_{ij} \otimes E_{ji} + v E_{ji} \otimes E_{ij} \right),
\end{aligned}
\tag{2.1}
$$

where $(E_{ij})_{lk} = \delta_{il} \delta_{jk}$, $i, j, l, k = 1, \ldots, m$ are elementary unit matrices and the rational functions $f(u, v)$ and $g(u, v)$ are

$$
f(u, v) = \frac{qu - q^{-1}v}{u - v}, \quad g(u, v) = \frac{q - q^{-1}}{u - v},
\tag{2.2}
$$

with $q$ a complex parameter not equal to zero. This matrix acts in the tensor product $\mathbf{C}^m \otimes \mathbf{C}^m$ and defines commutation relations

$$
R(u, v) \left( T(u) \otimes \mathbf{1} \right) \left( \mathbf{1} \otimes T(v) \right) = \left( \mathbf{1} \otimes T(v) \right) \left( T(u) \otimes \mathbf{1} \right) R(u, v)
\tag{2.3}
$$

for the quantum monodromy matrix $T(u)$ of some quantum integrable model.

Equation (2.3) holds in the tensor product $\mathbf{C}^m \otimes \mathbf{C}^m \otimes \mathscr{H}$, where $\mathscr{H}$ is a Hilbert space of the model. Being projected onto specific matrix element the commutation relation (2.3) can be written as the relation for the monodromy matrix elements acting in the Hilbert space $\mathscr{H}$

$$
\begin{aligned}
[T_{i,j}(u), T_{k,l}(v)] = {} & \left( f(u, v) - 1 \right) \left\{ \delta_{lj} T_{k,j}(v) T_{i,l}(u) - \delta_{ik} T_{k,j}(u) T_{i,l}(v) \right\} \\
& + g(u, v) \left\{ \left( u \delta_{l<j} + v \delta_{j<l} \right) T_{k,j}(v) T_{i,l}(u) - \left( u \delta_{i<k} + v \delta_{k<i} \right) T_{k,j}(u) T_{i,l}(v) \right\},
\end{aligned}
\tag{2.4}
$$

where $\delta_{i<j} = 1$ if $i < j$ and 0 otherwise.

The transfer matrix is defined as the trace of the monodromy matrix

$$
\mathscr{T}(u) = \operatorname{tr} T(u) = \sum_{j=1}^{m} T_{j,j}(u).
\tag{2.5}
$$

It follows from the $RTT$-relation (2.3) that $[\mathscr{T}(u), \mathscr{T}(v)] = 0$. Thus the transfer matrix can be used as a generating function of integrals of motion of an integrable system.

We call such a model $U_q(\widehat{\mathfrak{gl}}_m)$ based quantum integrable model because of the $R$-matrix used in definition of the commutation relations (2.3) and also because the centerless quantum affine algebra $U_q(\widehat{\mathfrak{gl}}_m)$ itself can be defined using the commutation relations (2.3) by identification of the quantum monodromy matrix $T(u)$ with the generating series of the Borel subalgebra elements in $U_q(\widehat{\mathfrak{gl}}_m)$.

Assume that the operator

$$
\mathscr{L} = \lim_{u \to \infty} T(u) \quad \text{with} \quad \mathscr{L} = \sum_{i,j=1}^{m} E_{ij} \otimes \mathscr{L}_{i,j}
$$

is well defined. We call such operators $\mathscr{L}_{i,j}$ zero modes operators[1] and it follows from the commutation relations (2.4) that[2]

$$
\mathscr{L}_{i,i} T_{k,l}(u) = q^{\delta_{il} - \delta_{ik}} T_{k,l}(u) \mathscr{L}_{i,i}.
\tag{2.6}
$$

---

[1] In fact the zero mode generators exist whatever is the asymptotic behavior of $T(u)$ at $u = \infty$. We have taken this particular behavior to simplify the presentation.

[2] To get this result one needs to assume that the zero mode matrix $\mathscr{L}$ is upper-triangular.

Matrix elements $T_{i,j}(u)$ of the monodromy matrix $T(u)$ form the algebra with the commutation relations (2.4) which we denote as $\mathscr{A}_m^q$. Further on we will consider certain morphisms which relate algebras $\mathscr{A}_m^q$ and $\mathscr{A}_m^{q^{-1}}$ (see section 3) as well as embeddings of the smaller rank algebra $\mathscr{A}_{m-1}^q$ into the bigger rank algebra $\mathscr{A}_m^q$.

We wish here to make some comments on the distinction between $\mathscr{A}_m^q$ and $U_q(\widehat{\mathfrak{gl}}_m)$ algebras. The $R$-matrix we use is definitely the one associated to the $U_q(\widehat{\mathfrak{gl}}_m)$ algebra. However, in order to define this algebra, more elements are needed, such as the Lax operator(s) and their expansion with respect to the spectral parameter. On the other hand, the definition of an integrable model 'only' needs a monodromy matrix obeying an $RTT$-relation. Hence, we refer to the $\mathscr{A}_m^q$ algebra when dealing with this monodromy matrix, while the denomination $U_q(\widehat{\mathfrak{gl}}_m)$ will be used when mentioning the underlying models.

Most of the time, one may identify the $\mathscr{A}_m^q$ algebra with a Borel subalgebra in the quantum affine algebra $U_q(\widehat{\mathfrak{gl}}_m)$. This allows to define the model and its Bethe vectors. However, when considering dual Bethe vectors and the morphism $\Psi$ (see section 3.2) the situation is more delicate. This is particularly acute when the central charge is not zero, and we use the $\mathscr{A}_m^q$ algebra to bypass these subtleties. In particular, the morphism $\Psi$ maps $\mathscr{A}_m^q$ to $\mathscr{A}_m^{q^{-1}}$, while it maps $U_q^+$ to $U_{q^{-1}}^-$, where $U_q^+$ and $U_q^-$ are dual Borel subalgebras in $U_q(\widehat{\mathfrak{gl}}_m)$.

A similar discussion can be found in [32] on the Yangian case.

## 2.2 Notation

In this paper we use notation and conventions of the work [58]. Besides the functions $g(u,v)$ and $f(u,v)$ (2.2), we introduce the rational functions

$$g^{(r)}(u,v) = v\, g(u,v), \quad g^{(l)}(u,v) = u\, g(u,v). \tag{2.7}$$

Let us formulate now a convention on the notation. We denote sets of variables by bar, for example, $\bar{u}$. When dealing with several of them, we may equip these sets or subsets with additional superscript: $\bar{s}^i$, $\bar{t}^\nu$, etc. Individual elements of the sets or subsets are denoted by Latin subscripts, for instance, $u_j$ is an element of $\bar{u}$, $t_k^i$ is an element of $\bar{t}^i$ etc. Subsets complementary to the elements $u_j$ (resp. $t_k^i$) are denoted by bar, i.e. $\bar{u}_j$ (resp. $\bar{t}_k^i$). Thus, $\bar{u}_j = \bar{u} \setminus \{u_j\}$ and $\bar{t}_k^i = \bar{t}^i \setminus \{t_k^i\}$. For any set $\bar{u}$, we will note $\#\bar{u}$ the cardinality of the set $\bar{u}$. As a rule, the number of elements in the sets is not shown explicitly in the equations, however we give these cardinalities in special comments to the formulas.

We use a shorthand notation for products of functions $f$, $g$ or $g^{(l,r)}$: if some function depends on a set of variables (or two sets of variables), this means that one should take the product over the corresponding set (or double product over the two sets). For example,

$$g^{(l)}(\bar{u},v) = \prod_{u_j \in \bar{u}} g^{(l)}(u_j,v), \quad f(\bar{t}_j^\mu, t_j^\mu) = \prod_{\substack{t_\ell^\mu \in \bar{t}^\mu \\ \ell \neq j}} f(t_\ell^\mu, t_j^\mu), \quad f(\bar{s}^j, \bar{t}^i) = \prod_{s_k^j \in \bar{s}^j} \prod_{t_\ell^i \in \bar{t}^i} f(s_k^j, t_\ell^i). \tag{2.8}$$

The same convention is applied to the products of commuting operators. Note that (2.4) implies in particular that

$$[T_{i,j}(u), T_{i,j}(v)] = 0, \quad \forall\, i,j = 1,\dots,m. \tag{2.9}$$

Thus, the notation

$$T_{i,j}(\bar{u}) = \prod_{u_k \in \bar{u}} T_{i,j}(u_k) \tag{2.10}$$

is well defined.

By definition, any product over the empty set is equal to 1. A double product is equal to 1 if at least one of the sets is empty. Below we will extend this convention to the products of eigenvalues of the diagonal monodromy matrix entries and their ratios (see (3.3)).

# 3 Bethe vectors

**Pseudovacuum vector.** The entries $T_{i,j}(u)$ of the monodromy matrix $T(u)$ act in a Hilbert space $\mathscr{H}$. We do not specify $\mathscr{H}$, but we assume that it contains a *pseudovacuum vector* $|0\rangle$, such that

$$
\begin{aligned}
T_{i,i}(u)|0\rangle &= \lambda_i(u)|0\rangle, & i &= 1,\dots,m, \\
T_{i,j}(u)|0\rangle &= 0, & i &> j,
\end{aligned}
\tag{3.1}
$$

where $\lambda_i(u)$ are some scalar functions. In the framework of the generalized model [13] considered in this paper, the scalar functions $\lambda_i(u)$ remain free functional parameters. Let us briefly recall that the generalized model is a class of models possessing the same $R$-matrix (2.1) and having a pseudovacuum vector with the properties (3.1) (see [13, 58] for more details). Any representative of this class can be characterized by a set of functional parameters that are the ratios of the vacuum eigenvalues $\lambda_i$:

$$
\alpha_i(u) = \frac{\lambda_i(u)}{\lambda_{i+1}(u)}, \qquad i = 1,\dots,m-1.
\tag{3.2}
$$

We extend to these functions the convention on the shorthand notation (2.8), for instance:

$$
\lambda_k(\bar{u}) = \prod_{u_j \in \bar{u}} \lambda_k(u_j), \qquad \alpha_i(\bar{t}^i) = \prod_{t_\ell^i \in \bar{t}^i} \alpha_i(t_\ell^i).
\tag{3.3}
$$

**Coloring.** In physical models, the space $\mathscr{H}$ is generated by states with quasiparticles of different types (colors). In $U_q(\widehat{\mathfrak{gl}}_m)$ based models quasiparticles may have $N = m-1$ colors. For any set $\{r_1,\dots,r_N\}$ of non-negative integers, we say that a state has coloring $\{r_1,\dots,r_N\}$, if it contains $r_i$ quasiparticles of the color $i$. This definition can be formalized at the level of the quantum algebra $U_q(\widehat{\mathfrak{gl}}_m)$ through the diagonal zero modes operators $\mathscr{L}_{k,k}$ (2.6). The colors correspond to the eigenvalues under the commuting generators[3]

$$
\mathfrak{h}_j = \prod_{k=1}^{j} \mathscr{L}_{k,k}, \quad j = 1,\dots,m-1.
\tag{3.4}
$$

Indeed, one can check from (2.6) that

$$
\mathfrak{h}_j\, T_{k,l}(z) = q^{\varepsilon_j(k,l)}\, T_{k,l}(z)\, \mathfrak{h}_j \quad \text{with} \quad
\begin{cases}
\varepsilon_j(k,l) = -1, & \text{if} \quad k \le j < l, \\
\varepsilon_j(k,l) = +1, & \text{if} \quad l \le j < k, \\
\varepsilon_j(k,l) = 0 & \text{otherwise.}
\end{cases}
\tag{3.5}
$$

The eigenvalues $\varepsilon_j(k,l)$ just correspond to the coloring mentioned above.

To get a zero coloring of the vector $|0\rangle$, one needs to shift $\mathfrak{h}_j$ to $h_j = \mathfrak{h}_j \prod_{k=1}^{j} \lambda_k[0]^{-1}$, where $\lambda_k[0]$ is the eigenvalue of $|0\rangle$ under $\mathscr{L}_{k,k}$. Then, all states in $\mathscr{H}$ have positive (or null) colors. A state with a given coloring can be obtained by successive application of the creation operators $T_{i,j}$ with $i < j$ to the vector $|0\rangle$. Acting on a state, an operator $T_{i,j}$ with $i < j$ adds one quasiparticle of each colors $i,\dots,j-1$. In particular, the operator $T_{i,i+1}$ creates one quasiparticle of the color $i$, the operator $T_{1,m}$ creates $N$ quasiparticles of $N$ different colors. The diagonal operators $T_{i,i}$ are neutral, the matrix elements $T_{i,j}$ with $i > j$ play the role of annihilation operators. They remove from any state the quasiparticles with the colors $j,\dots,i-1$, one particle of each color. In particular, if $j-1 < k < i$, and the annihilation operator $T_{i,j}$ acts on a state in which there are no particles of the color $k$, then its action yields zero.

---

[3]The last generator $\mathfrak{h}_m$ is central, see (3.5).

**Bethe vectors.** Bethe vectors belong to the space $\mathscr{H}$. Their distinctive feature is that when Bethe equations are fulfilled (see section 3.3) they become eigenvectors of the transfer matrix (2.5). Several explicit forms for Bethe vectors can be found in [37]. We do not use them in the present paper, however, in section 4.1 we give a recursion that formally allows the Bethe vectors to be explicitly constructed. In the present section, we only fix their normalization.

Generically, Bethe vectors are certain polynomials in the creation operators $T_{i,j}$ applied to the vector $|0\rangle$. These polynomials are eigenvectors under the Cartan generators $\mathscr{L}_{k,k}$, and hence they are also eigenvectors of the color generators $h_j$. Thus, Bethe vectors have a definite coloring and contain only terms with the same coloring.

A generic Bethe vector of $U_q(\widehat{\mathfrak{gl}}_m)$ based model depends on $N = m-1$ sets of variables $\bar{t}^1, \bar{t}^2, \ldots, \bar{t}^N$ called Bethe parameters. We denote Bethe vectors by $\mathbb{B}(\bar{t})$, where

$$\bar{t} = \{t_1^1, \ldots, t_{r_1}^1; t_1^2, \ldots, t_{r_2}^2; \ldots; t_1^N, \ldots, t_{r_N}^N\}, \tag{3.6}$$

and the cardinalities $r_i$ of the sets $\bar{t}^i$ coincide with the coloring. Thus, each Bethe parameter $t_k^i$ can be associated with a quasiparticle of the color $i$.

Bethe vectors are symmetric over permutations of the parameters $t_k^i$ within the set $\bar{t}^i$ (see e.g. [37]). However, they are not symmetric over permutations over parameters belonging to different sets $\bar{t}^i$ and $\bar{t}^j$.

We have already mentioned that a generic Bethe vector has the form of a polynomial in $T_{i,j}$ with $i < j$ applied to the pseudovacuum $|0\rangle$. Among all the terms of this polynomial, there is one monomial that contains the operators $T_{i,j}$ with $j - i = 1$ only. Let us call this term the *main term* and denote it by $\widetilde{\mathbb{B}}(\bar{t})$. Then

$$\mathbb{B}(\bar{t}) = \widetilde{\mathbb{B}}(\bar{t}) + \ldots, \tag{3.7}$$

where the ellipsis stands for all the terms with the same coloring that contain at least one operator $T_{i,j}$ with $j - i > 1$. We fix the normalization of the Bethe vectors by requiring the following form of the main term

$$\widetilde{\mathbb{B}}(\bar{t}) = \frac{T_{1,2}(\bar{t}^1) \ldots T_{N,N+1}(\bar{t}^N) |0\rangle}{\prod_{i=1}^{N} \lambda_{i+1}(\bar{t}^i) \prod_{i=1}^{N-1} f(\bar{t}^{i+1}, \bar{t}^i)}. \tag{3.8}$$

Recall that we use here the shorthand notation for the products of the functions $\lambda_{j+1}$ and $f$, as well as for a set of commuting operators $T_{i,i+1}$. Let us stress that this normalization is different from the one used in [37] where the coefficient of the operator product in the definition of $\widetilde{\mathbb{B}}(\bar{t})$ was just 1. This additional normalization factor is convenient, in particular because the scalar products of the Bethe vectors depend on the ratios $\alpha_i$ (3.2) only.

Since the operators $T_{i,i+1}$ and $T_{j,j+1}$ do not commute for $i \neq j$, the main term can be written in several forms corresponding to different ordering of the monodromy matrix entries. The ordering in (3.8) naturally arises if we construct Bethe vectors via the nesting procedure corresponding to the embedding of $\mathscr{A}_{m-1}^q$ in $\mathscr{A}_m^q$ to the lower-right corner of the monodromy matrix $T(u)$.

## 3.1 Morphism of Bethe vectors

The quantum algebras $\mathscr{A}_m^q$ and $\mathscr{A}_m^{q^{-1}}$ are related by a morphism $\varphi$ [37]:

$$\varphi\big(T(u)\big) = U \, \widetilde{T}^t(u) \, U^{-1}, \quad \text{i.e.} \quad \varphi\big(T_{a,b}(u)\big) = \widetilde{T}_{m+1-b, m+1-a}(u), \tag{3.9}$$

where $U = \sum_{i=1}^{m} E_{i,m+1-i}$ and we put a tilde on the generators of $\mathscr{A}_m^{q^{-1}}$ to distinguish them from those of $\mathscr{A}_m^q$. $\varphi$ defines an idempotent isomorphism from $\mathscr{A}_m^q$ to $\mathscr{A}_m^{q^{-1}}$. This mapping

also acts on the vacuum eigenvalues $\lambda_i(u)$ (3.1) and their ratios $\alpha_i(u)$ (3.2)

$$\varphi: \begin{cases} \lambda_i(u) & \to \ \widetilde{\lambda}_{m+1-i}(u), \qquad i = 1, \ldots, m, \\ \alpha_i(u) & \to \ \frac{1}{\widetilde{\alpha}_{m-i}(u)}, \qquad i = 1, \ldots, m-1. \end{cases} \tag{3.10}$$

We can extend this morphism to representations, defining $\varphi(|0\rangle) = \widetilde{|0\rangle}$, where $|0\rangle$ and $\widetilde{|0\rangle}$ are the pseudovacua in $\mathscr{H}$ and $\widetilde{\mathscr{H}}$ respectively. It has been shown in [37] that this morphism induces the following correspondence between Bethe vectors

**Lemma 3.1.** *The morphism $\varphi$ induces a mapping of Bethe vectors $\mathbb{B}_q(\bar{t}) \in \mathscr{H}$ to Bethe vectors $\mathbb{B}_{q^{-1}}(\bar{t}) \in \widetilde{\mathscr{H}}$:*

$$\varphi\left(\mathbb{B}_q(\vec{t})\right) = \frac{\mathbb{B}_{q^{-1}}(\overleftarrow{t})}{\prod_{k=1}^{N} \widetilde{\alpha}_{N+1-k}(\bar{t}^k)}, \tag{3.11}$$

*where we have introduced the special orderings of the sets of Bethe parameters*[4]

$$\vec{t} = \{\bar{t}^1, \bar{t}^2, \ldots, \bar{t}^N\} \qquad and \qquad \overleftarrow{t} = \{\bar{t}^N, \ldots, \bar{t}^2, \bar{t}^1\}. \tag{3.12}$$

### 3.2 Dual Bethe vectors

Dual Bethe vectors belong to the dual Hilbert space $\mathscr{H}^*$, and they are polynomials in $T_{i,j}$ with $i > j$ applied from the right to the dual pseudovacuum vector $\langle 0|$. This vector possesses the properties similar to (3.1)

$$\begin{aligned} \langle 0|T_{i,i}(u) &= \lambda_i(u)\langle 0|, & i &= 1, \ldots, m, \\ \langle 0|T_{i,j}(u) &= 0, & i &< j, \end{aligned} \tag{3.13}$$

where the functions $\lambda_i(u)$ are the same as in (3.1).

We denote dual Bethe vectors by $\mathbb{C}(\bar{t})$, where the set of Bethe parameters $\bar{t}$ consists of several sets $\bar{t}^i$ as in (3.6). As it was done for Bethe vectors, we can introduce the coloring of the dual Bethe vectors, with now the role of creation and annihilation operators reversed.

One can obtain dual Bethe vectors via the special antimorphism $\Psi$ given by

$$\Psi\big(T(u)\big) = \widetilde{T}^t(u^{-1}), \quad \text{i.e.} \quad \Psi\big(T_{a,b}(u)\big) = \widetilde{T}_{b,a}(u^{-1}). \tag{3.14}$$

$\Psi$ defines an idempotent antimorphism from $\mathscr{A}_m^q$ to $\mathscr{A}_m^{q^{-1}}$. Let us extend the action of this antimorphism to the pseudovacuum vectors by

$$\begin{aligned} \Psi(|0\rangle) &= \widetilde{\langle 0|}, & \Psi(A|0\rangle) &= \widetilde{\langle 0|}\Psi(A), \\ \Psi(\langle 0|) &= \widetilde{|0\rangle}, & \Psi(\langle 0|A) &= \Psi(A)\widetilde{|0\rangle}, \end{aligned} \tag{3.15}$$

where $A$ is any product of $T_{i,j}$. Then it turns out that [37]

$$\Psi\big(\mathbb{B}_q(\bar{t})\big) = \mathbb{C}_{q^{-1}}(\bar{t}^{-1}), \qquad \Psi\big(\mathbb{C}_q(\bar{t})\big) = \mathbb{B}_{q^{-1}}(\bar{t}^{-1}), \tag{3.16}$$

where, again, we put a subscript on (dual) Bethe vectors to distinguish the ones of $\mathscr{A}_m^q$ from those of $\mathscr{A}_m^{q^{-1}}$. We used the notation

$$\bar{t}^{-1} \equiv \frac{1}{\bar{t}} \equiv \left\{ \frac{1}{t_1^1}, \frac{1}{t_2^1}, \ldots, \frac{1}{t_{r_1}^1}, \frac{1}{t_1^2}, \ldots, \frac{1}{t_{r_N}^N} \right\}.$$

---

[4]Let us stress that the order of the Bethe parameters within every subset $\bar{t}^k$ is not essential.

The main term of the dual Bethe vector can be obtained from (3.8) via the mapping[5] $\Psi$:

$$\widetilde{\mathbb{C}}(\bar{t}) = \frac{\langle 0| T_{N+1,N}(\bar{t}^N) \dots T_{2,1}(\bar{t}^1)}{\prod_{i=1}^{N} \lambda_{i+1}(\bar{t}^i) \prod_{i=1}^{N-1} f(\bar{t}^{i+1}, \bar{t}^i)}. \tag{3.17}$$

Finally, using the morphism $\varphi$ we obtain a relation between dual Bethe vectors corresponding to the quantum algebras $\mathscr{A}_m^q$ and $\mathscr{A}_m^{q^{-1}}$

$$\varphi\left(\mathbb{C}_q(\vec{t})\right) = \frac{\mathbb{C}_{q^{-1}}(\vec{t})}{\prod_{k=1}^{N} \widetilde{\alpha}_{N+1-k}(\bar{t}^k)}. \tag{3.18}$$

## 3.3 On-shell Bethe vectors

For generic Bethe vectors, the Bethe parameters $t_k^i$ are generic complex numbers. If these parameters satisfy a special system of equations (the Bethe equations, see (3.19)), then the corresponding vector becomes an eigenvector of the transfer matrix (2.5). In this case it is called *on-shell Bethe vector*. In most of the paper we consider generic Bethe vectors. However, for the calculation of the norm of Bethe vectors we will consider on-shell Bethe vectors. In that case, the parameters $\bar{t}$ and $\alpha_\mu$ will be related by the following system of Bethe equations

$$\alpha_\nu(t_j^\nu) = \frac{f(t_j^\nu, \bar{t}_j^\nu) f(\bar{t}^{\nu+1}, t_j^\nu)}{f(\bar{t}_j^\nu, t_j^\nu) f(t_j^\nu, \bar{t}^{\nu-1})}, \qquad \nu = 1, \dots, N, \quad j = 1, \dots, r_\nu, \tag{3.19}$$

and we recall that $\bar{t}_j^\nu = \bar{t}^\nu \setminus \{t_j^\nu\}$. Usually, when the functions $\alpha_\mu$ are given (and define a physical model), one considers these equations as a way to determine the allowed values for the Bethe parameters $\bar{t}$. For the generalized models, where the functions $\alpha_\mu$ are not fixed, the Bethe equations form a set of relations between the functional parameters $\alpha_\mu(t_j^\mu)$ and the Bethe parameters $t_k^\nu$.

## 3.4 Coproduct property and composite models

The proofs for the results shown in the present paper rely on a coproduct property for Bethe vectors, which connects the Bethe vectors belonging to the spaces $\mathscr{H}^{(1)}$ and $\mathscr{H}^{(2)}$ to the Bethe vectors in the space $\mathscr{H}^{(1)} \otimes \mathscr{H}^{(2)}$. This property is intimately related to the notion of composite model, that we introduce now. It is important to point out that in this section we consider Bethe vectors corresponding to different monodromy matrices. We stress it by adding the monodromy matrix to the list of the Bethe vectors arguments. Namely, the notation $\mathbb{B}(\bar{t}|T)$ means that the Bethe vector $\mathbb{B}(\bar{t})$ corresponds to the monodromy matrix $T$.

In a composite model, the monodromy matrix $T(u)$ is presented as a product of two partial monodromy matrices [32, 62–64]:

$$T(u) = T^{(2)}(u) T^{(1)}(u). \tag{3.20}$$

Here every $T^{(l)}(u)$ satisfies the $RTT$-relation (2.3) and has its own pseudovacuum vector $|0\rangle^{(l)}$ and dual vector $\langle 0|^{(l)}$, such that $|0\rangle = |0\rangle^{(1)} \otimes |0\rangle^{(2)}$ and $\langle 0| = \langle 0|^{(1)} \otimes \langle 0|^{(2)}$. The operators $T_{i,j}^{(2)}(u)$ and $T_{k,l}^{(1)}(v)$ act in different spaces, and hence, they commute with each other. We assume that

$$\begin{aligned} T_{i,i}^{(l)}(u)|0\rangle^{(l)} &= \lambda_i^{(l)}(u)|0\rangle^{(l)}, \\ \langle 0|^{(l)} T_{i,i}^{(l)}(u) &= \lambda_i^{(l)}(u)\langle 0|^{(l)}, \end{aligned} \qquad i = 1, \dots, m, \qquad l = 1, 2, \tag{3.21}$$

---

[5]To get a dual Bethe vector in $U_q(\widehat{\mathfrak{gl}}_m)$ one should start from $U_{q^{-1}}(\widehat{\mathfrak{gl}}_m)$, see [37] where these considerations are detailed.

where $\lambda_i^{(l)}(u)$ are new free functional parameters. We also introduce

$$\alpha_k^{(l)}(u) = \frac{\lambda_k^{(l)}(u)}{\lambda_{k+1}^{(l)}(u)}, \qquad l = 1, 2, \qquad k = 1, \ldots, N. \tag{3.22}$$

Obviously

$$\lambda_i(u) = \lambda_i^{(1)}(u)\lambda_i^{(2)}(u), \qquad \alpha_k(u) = \alpha_k^{(1)}(u)\alpha_k^{(2)}(u). \tag{3.23}$$

The partial monodromy matrices $T^{(l)}(u)$ have the corresponding Bethe vectors $\mathbb{B}(\bar{t}|T^{(l)})$ and dual Bethe vectors $\mathbb{C}(\bar{s}|T^{(l)})$. A Bethe vector $\mathbb{B}(\bar{t}|T)$ of the total monodromy matrix $T(u)$ can be expressed in terms partial Bethe vectors $\mathbb{B}(\bar{t}|T^{(l)})$ via coproduct formula [34, 35]

$$\mathbb{B}(\bar{t}|T) = \sum \frac{\prod_{\nu=1}^N \alpha_\nu^{(2)}(\bar{t}_{\text{i}}^\nu) f(\bar{t}_{\text{ii}}^\nu, \bar{t}_{\text{i}}^\nu)}{\prod_{\nu=1}^{N-1} f(\bar{t}_{\text{ii}}^{\nu+1}, \bar{t}_{\text{i}}^\nu)} \, \mathbb{B}(\bar{t}_{\text{i}}|T^{(1)}) \otimes \mathbb{B}(\bar{t}_{\text{ii}}|T^{(2)}). \tag{3.24}$$

Here all the sets of the Bethe parameters $\bar{t}^\nu$ are divided into two subsets $\bar{t}^\nu \Rightarrow \{\bar{t}_{\text{i}}^\nu, \bar{t}_{\text{ii}}^\nu\}$, and the sum is taken over all possible partitions.

A similar formula exists for the dual Bethe vectors $\mathbb{C}(\bar{s}|T)$ (see appendix C)

$$\mathbb{C}(\bar{s}|T) = \sum \frac{\prod_{\nu=1}^N \alpha_\nu^{(1)}(\bar{s}_{\text{ii}}^\nu) f(\bar{s}_{\text{i}}^\nu, \bar{s}_{\text{ii}}^\nu)}{\prod_{\nu=1}^{N-1} f(\bar{s}_{\text{i}}^{\nu+1}, \bar{s}_{\text{ii}}^\nu)} \, \mathbb{C}(\bar{s}_{\text{ii}}|T^{(2)}) \otimes \mathbb{C}(\bar{s}_{\text{i}}|T^{(1)}), \tag{3.25}$$

where the sum is organised in the same way as in (3.24).

## 4 Main results

In this section we present the main results of the paper. For generic Bethe vectors, we provide recursion formulas (section 4.1), sum formulas for their scalar products (section 4.2), and recursions for the highest coefficients (section 4.3). For on-shell Bethe vectors, we exhibit a Gaudin determinant form for their norm (section 4.4).

We would like to stress that all the results are given in terms of rational functions $f(u, v)$ (2.2), $g^{(l,r)}(u, v)$ (2.7), and ratios of the eigenvalues $\alpha_i(u)$ (3.2). Therefore, they can easily be compared with the results obtained in [58, 59] for the models with the Yangian $R$-matrix. This comparison shows that in both cases the results have completely the same structure. The only slight difference consists in the fact that in the case of the Yangian the functions $g^{(l)}(u, v)$ and $g^{(r)}(u, v)$ degenerate into one function $g(u, v)$. As we have already mentioned in Introduction, this similarity of the results is not accidental. It is explained by the similarity of the corresponding $R$-matrices. Due to this reason the proofs of most of the results listed above for the $U_q(\widehat{\mathfrak{gl}}_m)$ based models are identical to the corresponding proofs in the Yangian case. To show this we give a detailed proof of the sum formula (4.11). However, for the proofs of other statements we refer the reader to the works [58, 59].

The essential difference between models that are described by $Y(\mathfrak{gl}_m)$ and $U_q(\widehat{\mathfrak{gl}}_m)$ algebras is the action of morphisms $\varphi$ (3.9) and $\Psi$ (3.14). In particular, in the case of the Yangian, the antimorphism (3.14) turns into an endomorphism, while in the $U_q(\widehat{\mathfrak{gl}}_m)$ case this mapping connects two different algebras. Therefore, all the proofs based on the application of the mappings $\varphi$ and $\Psi$, are given in details.

### 4.1 Recursion for Bethe vectors

Here we give recursions for (dual) Bethe vectors. The corresponding proofs are given in section 5.

**Proposition 4.1.** *Bethe vectors of $U_q(\widehat{\mathfrak{gl}}_m)$ based models satisfy a recursion*

$$\mathbb{B}(\{z,\bar{t}^1\};\{\bar{t}^k\}_2^N) = \sum_{j=2}^{N+1} \frac{T_{1,j}(z)}{\lambda_2(z)} \sum_{\text{part}(\bar{t}^2,\ldots,\bar{t}^{j-1})} \mathbb{B}(\{\bar{t}^1\};\{\bar{t}_{\mathrm{I\!I}}^k\}_2^{j-1};\{\bar{t}^k\}_j^N)$$

$$\times \frac{\prod_{\nu=2}^{j-1}\alpha_\nu(\bar{t}_{\mathrm{I}}^\nu)\,g^{(l)}(\bar{t}_{\mathrm{I}}^\nu,\bar{t}_{\mathrm{I}}^{\nu-1})f(\bar{t}_{\mathrm{I\!I}}^\nu,\bar{t}_{\mathrm{I}}^\nu)}{\prod_{\nu=1}^{j-1} f(\bar{t}^{\nu+1},\bar{t}_{\mathrm{I}}^\nu)}. \quad (4.1)$$

*Here for $j > 2$ the sets of Bethe parameters $\bar{t}^2,\ldots,\bar{t}^{j-1}$ are divided into disjoint subsets $\bar{t}_{\mathrm{I}}^\nu$ and $\bar{t}_{\mathrm{I\!I}}^\nu$ ($\nu = 2,\ldots,j-1$) such that the subset $\bar{t}_{\mathrm{I}}^\nu$ consists of one element only: $\#\bar{t}_{\mathrm{I}}^\nu = 1$. The sum is taken over all partitions of this type. We set $\bar{t}_{\mathrm{I}}^1 \equiv z$ and $\bar{t}^{N+1} = \emptyset$. Recall also that $N = m-1$.*

We used the following notation in proposition 4.1

$$\begin{aligned}
\mathbb{B}(\{z,\bar{t}^1\};\{\bar{t}^k\}_2^N) &= \mathbb{B}(\{z,\bar{t}^1\};\bar{t}^2;\ldots;\bar{t}^N),\\
\mathbb{B}(\{\bar{t}^1\};\{\bar{t}_{\mathrm{I\!I}}^k\}_2^{j-1};\{\bar{t}^k\}_j^N) &= \mathbb{B}(\bar{t}^1;\bar{t}_{\mathrm{I\!I}}^2;\ldots;\bar{t}_{\mathrm{I\!I}}^{j-1};\bar{t}^j;\ldots;\bar{t}^N).
\end{aligned} \quad (4.2)$$

Similar notation will be used throughout the paper.

*Remark.* We stress that each of the subsets $\bar{t}_{\mathrm{I}}^2,\ldots,\bar{t}_{\mathrm{I}}^N$ in (4.1) must consist of exactly one element. However, this condition cannot be achieved if the original Bethe vector $\mathbb{B}(t)$ contains an empty set $\bar{t}^k = \emptyset$ for some $k \in [2,\ldots,N]$. In this case, the sum over $j$ in (4.1) ends at $j = k$. If $\mathbb{B}(t)$ contains several empty sets $\bar{t}^{k_1},\ldots,\bar{t}^{k_\ell}$, then the sum finishes at $j = \min(k_1,\ldots,k_\ell)$.

Using the mapping (3.9) one can obtain a second recursion for the Bethe vectors:

**Proposition 4.2.** *Bethe vectors of $U_q(\widehat{\mathfrak{gl}}_m)$ based models satisfy a recursion*

$$\mathbb{B}(\{\bar{t}^k\}_1^{N-1};\{z,\bar{t}^N\}) = \sum_{j=1}^{N} \frac{T_{j,N+1}(z)}{\lambda_{N+1}(z)} \sum_{\text{part}(\bar{t}^j,\ldots,\bar{t}^{N-1})} \mathbb{B}(\{\bar{t}^k\}_1^{j-1};\{\bar{t}_{\mathrm{I\!I}}^k\}_j^{N-1};\bar{t}^N)$$

$$\times \frac{\prod_{\nu=j}^{N-1} g^{(r)}(\bar{t}_{\mathrm{I}}^{\nu+1},\bar{t}_{\mathrm{I}}^\nu)f(\bar{t}_{\mathrm{I}}^\nu,\bar{t}_{\mathrm{I\!I}}^\nu)}{\prod_{\nu=j}^{N} f(\bar{t}_{\mathrm{I}}^\nu,\bar{t}^{\nu-1})}. \quad (4.3)$$

*Here for $j < N$ the sets of Bethe parameters $\bar{t}^j,\ldots,\bar{t}^{N-1}$ are divided into disjoint subsets $\bar{t}_{\mathrm{I}}^\nu$ and $\bar{t}_{\mathrm{I\!I}}^\nu$ ($\nu = j,\ldots,N-1$) such that the subset $\bar{t}_{\mathrm{I}}^\nu$ consists of one element: $\#\bar{t}_{\mathrm{I}}^\nu = 1$. The sum is taken over all partitions of this type. We set by definition $\bar{t}_{\mathrm{I}}^N \equiv z$ and $\bar{t}^0 = \emptyset$.*

*Remark.* If the Bethe vector $\mathbb{B}(t)$ contains several empty sets $\bar{t}^{k_1},\ldots,\bar{t}^{k_\ell}$, then the sum over $j$ in (4.3) begins with $j = \max(k_1,\ldots,k_\ell)+1$.

Acting with the antimorphism (3.14) onto equations (4.1) and (4.3) we arrive at

**Corollary 4.3.** *Dual Bethe vectors of $U_q(\widehat{\mathfrak{gl}}_m)$ based models satisfy recursions*

$$\mathbb{C}(\{z,\bar{s}^1\};\{\bar{s}^k\}_2^N) = \sum_{j=2}^{N+1} \sum_{\text{part}(\bar{s}^2,\ldots,\bar{s}^{j-1})} \mathbb{C}(\{\bar{s}^1\};\{\bar{s}_{\mathrm{I\!I}}^k\}_2^{j-1};\{\bar{s}^k\}_j^N) \frac{T_{j,1}(z)}{\lambda_2(z)}$$

$$\times \frac{\prod_{\nu=2}^{j-1}\alpha_\nu(\bar{s}_{\mathrm{I}}^\nu)\,g^{(r)}(\bar{s}_{\mathrm{I}}^\nu,\bar{s}_{\mathrm{I}}^{\nu-1})f(\bar{s}_{\mathrm{I\!I}}^\nu,\bar{s}_{\mathrm{I}}^\nu)}{\prod_{\nu=1}^{j-1} f(\bar{s}^{\nu+1},\bar{s}_{\mathrm{I}}^\nu)}, \quad (4.4)$$

*and*

$$\mathbb{C}(\{\bar{s}^k\}_1^{N-1};\{z,\bar{s}^N\}) = \sum_{j=1}^{N} \sum_{\text{part}(\bar{s}^j,\ldots,\bar{s}^{N-1})} \mathbb{C}(\{\bar{s}^k\}_1^{j-1};\{\bar{s}_{\mathrm{I\!I}}^k\}_j^{N-1};\bar{s}^N) \frac{T_{N+1,j}(z)}{\lambda_{N+1}(z)}$$

$$\times \frac{\prod_{\nu=j}^{N-1} g^{(l)}(\bar{s}_{\mathrm{I}}^{\nu+1},\bar{s}_{\mathrm{I}}^\nu)f(\bar{s}_{\mathrm{I}}^\nu,\bar{s}_{\mathrm{I\!I}}^\nu)}{\prod_{\nu=j}^{N} f(\bar{s}_{\mathrm{I}}^\nu,\bar{s}^{\nu-1})}. \quad (4.5)$$

*Here the summation over the partitions occurs as in the formulas* (4.1) *and* (4.3). *The subsets* $\bar{s}_{\mathrm{I}}^{\nu}$ *consist of one element:* $\#\bar{s}_{\mathrm{I}}^{\nu} = 1$. *If* $\mathbb{C}(\bar{s})$ *contains empty sets of Bethe parameters, then the sum cuts similarly to the case of the Bethe vectors* $\mathbb{B}(\bar{t})$. *By definition* $\bar{s}_{\mathrm{I}}^{1} \equiv z$ *in* (4.4), $\bar{s}_{\mathrm{I}}^{N} \equiv z$ *in* (4.5), *and* $\bar{s}^{0} = \bar{s}^{N+1} = \emptyset$.

Applying successively the recursion (4.1), we eventually express a Bethe vector with $\#\bar{t}^{1} = r_{1}$ as a linear combination of Bethe vectors with $\#\bar{t}^{1} = 0$. The latter effectively correspond to the quantum algebra $\mathscr{A}_{m-1}^{q}$:

$$\mathbb{B}^{(m)}(\emptyset; \{\bar{t}^{k}\}_{2}^{N}) = \mathbb{B}^{(m-1)}(\bar{t})\Big|_{\bar{t}^{k} \to \bar{t}^{k+1}}, \tag{4.6}$$

where we put a superscript to distinguish the Bethe vectors in $\mathscr{A}_{m}^{q}$ from those of $\mathscr{A}_{m-1}^{q}$. Thus, continuing this process we formally can reduce Bethe vectors of $\mathscr{A}_{m}^{q}$ to the known ones of $\mathscr{A}_{2}^{q}$. Similarly, one can build dual Bethe vectors via (4.4), (4.5). Unfortunately, these procedures are too cumbersome for explicit calculations. However, they can be used to prove various assertions by induction.

## 4.2 Sum formula for the scalar product

In this section we collect some results concerning scalar products of generic Bethe vectors. The proofs of propositions 4.4 and 4.5 literally coincide with the ones given in [58] for the Yangian case. Nevertheless, to illustrate this similarity we present one of these proofs (proposition 4.5) in section 6.

Let $\mathbb{B}(\bar{t})$ be a generic Bethe vector and $\mathbb{C}(\bar{s})$ be a generic dual Bethe vector. Then their scalar product is defined by

$$S(\bar{s}|\bar{t}) = \mathbb{C}(\bar{s})\mathbb{B}(\bar{t}). \tag{4.7}$$

Note that if $\#\bar{t}^{k} \neq \#\bar{s}^{k}$ for some $k \in \{1, \ldots, N\}$, then the scalar product vanishes. Indeed, in this case the numbers of creation and annihilation operators of the color $k$ in $\mathbb{B}(\bar{t})$ and $\mathbb{C}(\bar{s})$ respectively do not coincide. Thus, in the following we will assume that $\#\bar{t}^{k} = \#\bar{s}^{k} = r_{k}$, $k = 1, \ldots, N$.

Due to the normalizations (3.8) and (3.17), the scalar product of Bethe vectors depends on the functions $\lambda_{i}$ only through the ratios $\alpha_{i}$. The following proposition specifies this dependence.

**Proposition 4.4.** *Let* $\mathbb{B}(\bar{t})$ *be a generic Bethe vector and* $\mathbb{C}(\bar{s})$ *be a generic dual Bethe vector such that* $\#\bar{t}^{k} = \#\bar{s}^{k} = r_{k}$, $k = 1, \ldots, N$. *Then their scalar product is given by*

$$S(\bar{s}|\bar{t}) = \sum W_{\mathrm{part}}(\bar{s}_{\mathrm{I}}, \bar{s}_{\mathrm{II}}|\bar{t}_{\mathrm{I}}, \bar{t}_{\mathrm{II}}) \prod_{k=1}^{N} \alpha_{k}(\bar{s}_{\mathrm{I}}^{k})\alpha_{k}(\bar{t}_{\mathrm{II}}^{k}). \tag{4.8}$$

*Here all the sets of the Bethe parameters* $\bar{t}^{k}$ *and* $\bar{s}^{k}$ *are divided into two subsets* $\bar{t}^{k} \Rightarrow \{\bar{t}_{\mathrm{I}}^{k}, \bar{t}_{\mathrm{II}}^{k}\}$ *and* $\bar{s}^{k} \Rightarrow \{\bar{s}_{\mathrm{I}}^{k}, \bar{s}_{\mathrm{II}}^{k}\}$, *such that* $\#\bar{t}_{\mathrm{I}}^{k} = \#\bar{s}_{\mathrm{I}}^{k}$. *The sum is taken over all possible partitions of this type. The rational coefficients* $W_{\mathrm{part}}$ *depend on the partition of* $\bar{t}$ *and* $\bar{s}$, *but not on the vacuum eigenvalues* $\lambda_{k}$. *They are completely determined by the R-matrix of the model.*

Proposition 4.4 states that in the scalar product (4.7), the Bethe parameters of the type $k$ ($t_{j}^{k}$ or $s_{j}^{k}$) are arguments of the functions $\alpha_{k}$ only. This property has been proven for the case of Bethe vectors associated to the Yangian $Y(\mathfrak{gl}(m|n))$ in [58], and the proof for $\mathscr{A}_{m}^{q}$ follows exactly the same lines. The only difference lies in the relation (7.7) which now relates scalar products in different quantum algebras. However, this does not affect the functional

dependence stated in proposition 4.4. Simply, one has to work the proof simultaneously in $\mathscr{A}_m^q$ and in $\mathscr{A}_m^{q^{-1}}$. We refer the interested reader to [58] for more details.

We would like to stress that the rational functions $W_{\text{part}}$ are model independent. Thus, if two different models share the same $R$-matrix (2.1), then the scalar products of Bethe vectors in these models are given by (4.8) with the same coefficients $W_{\text{part}}$. In other words, the model dependent part of the scalar product entirely lies in the $\alpha_k$ functions.

The Highest Coefficient (HC) of the scalar product is defined as the rational coefficient corresponding to the partition $\bar{s}_{\mathrm{I}} = \bar{s}$, $\bar{t}_{\mathrm{I}} = \bar{t}$, and $\bar{s}_{\mathrm{II}} = \bar{t}_{\mathrm{II}} = \emptyset$. We denote the HC by $Z(\bar{s}|\bar{t})$:

$$W_{\text{part}}(\bar{s}, \emptyset | \bar{t}, \emptyset) = Z(\bar{s}|\bar{t}). \tag{4.9}$$

It corresponds to the coefficient of $\prod_{k=1}^{N} \alpha_k(\bar{s}^k)$ in the formula (4.8).

Similarly one can define a conjugated HC $\overline{Z}(\bar{s}|\bar{t})$ as the coefficient corresponding to the partition $\bar{s}_{\mathrm{II}} = \bar{s}$, $\bar{t}_{\mathrm{II}} = \bar{t}$, and $\bar{s}_{\mathrm{I}} = \bar{t}_{\mathrm{I}} = \emptyset$.

$$W_{\text{part}}(\emptyset, \bar{s}|\emptyset, \bar{t}) = \overline{Z}(\bar{s}|\bar{t}). \tag{4.10}$$

In the following, when speaking of both HC and conjugated HC, we will loosely call them *the HCs.*

The following proposition determines the general coefficient $W_{\text{part}}$ in terms of the HCs.

**Proposition 4.5.** *For a fixed partition $\bar{t}^k \Rightarrow \{\bar{t}_{\mathrm{I}}^k, \bar{t}_{\mathrm{II}}^k\}$ and $\bar{s}^k \Rightarrow \{\bar{s}_{\mathrm{I}}^k, \bar{s}_{\mathrm{II}}^k\}$ in (4.8) the rational coefficient $W_{\text{part}}$ has the following presentation in terms of the HCs:*

$$W_{\text{part}}(\bar{s}_{\mathrm{I}}, \bar{s}_{\mathrm{II}} | \bar{t}_{\mathrm{I}}, \bar{t}_{\mathrm{II}}) = Z(\bar{s}_{\mathrm{I}}|\bar{t}_{\mathrm{I}}) \, \overline{Z}(\bar{s}_{\mathrm{II}}|\bar{t}_{\mathrm{II}}) \, \frac{\prod_{k=1}^{N} f(\bar{s}_{\mathrm{II}}^k, \bar{s}_{\mathrm{I}}^k) f(\bar{t}_{\mathrm{I}}^k, \bar{t}_{\mathrm{II}}^k)}{\prod_{j=1}^{N-1} f(\bar{s}_{\mathrm{II}}^{j+1}, \bar{s}_{\mathrm{I}}^j) f(\bar{t}_{\mathrm{I}}^{j+1}, \bar{t}_{\mathrm{II}}^j)}. \tag{4.11}$$

Note that this proposition was already proven in the case of $\mathscr{A}_2^q$ in [13] and $\mathscr{A}_3^q$ in [48]. A comparison with the previous results obtained for $m = 3$ is given in appendix B. The proof for $\mathscr{A}_m^q$ is given in section 6.

## 4.3 Properties of the highest coefficient

In this section we list several useful properties of the HCs. Most of them are quite analogous to the properties of the HC in the Yangian case (see [58, 59]). The exception is the symmetry properties given in the following proposition.

**Proposition 4.6.** *The HC and conjugated HC in the quantum algebras $U_q(\widehat{\mathfrak{gl}}_m)$ and $U_{q^{-1}}(\widehat{\mathfrak{gl}}_m)$ are connected through the relations:*

$$Z_q(\vec{s}|\vec{t}) = \overline{Z}_{q^{-1}}(\vec{s}|\vec{t}), \tag{4.12}$$

$$\overline{Z}_q(\bar{s}|\bar{t}) = Z_{q^{-1}}(\bar{t}^{-1}|\bar{s}^{-1}), \tag{4.13}$$

*where again we put a subscript to indicate to which algebra the HC corresponds to.*

*The HC possesses also the symmetry*

$$Z_q(\vec{s}|\vec{t}) = Z_q(\overleftarrow{t}^{-1}|\overleftarrow{s}^{-1}). \tag{4.14}$$

The proof of this proposition is given in section 7.

Explicit expressions for the HC are known for $m = 2, 3$ [49, 60], but they become very ponderous when $m$ is generic. Fortunately, one can use relatively simple recursions described in the subsequent propositions.

**Proposition 4.7.** *The HC $Z(\bar{s}|\bar{t})$ possesses the following recursion over the set $\bar{s}^1$:*

$$Z(\bar{s}|\bar{t}) = \sum_{p=2}^{N+1} \sum_{\substack{\text{part}(\bar{s}^2,\ldots,\bar{s}^{p-1}) \\ \text{part}(\bar{t}^1,\ldots,\bar{t}^{p-1})}} \frac{g^{(l)}(\bar{t}_{\mathrm{I}}^1, \bar{s}_{\mathrm{I}}^1) f(\bar{t}_{\mathrm{I}}^1, \bar{t}_{\mathrm{II}}^1) f(\bar{t}_{\mathrm{II}}^1, \bar{s}_{\mathrm{I}}^1)}{f(\bar{s}^p, \bar{s}_{\mathrm{I}}^{p-1})}$$

$$\times \prod_{\nu=2}^{p-1} \frac{g^{(r)}(\bar{s}_{\mathrm{I}}^{\nu}, \bar{s}_{\mathrm{I}}^{\nu-1}) g^{(l)}(\bar{t}_{\mathrm{I}}^{\nu}, \bar{t}_{\mathrm{I}}^{\nu-1}) f(\bar{s}_{\mathrm{II}}^{\nu}, \bar{s}_{\mathrm{I}}^{\nu}) f(\bar{t}_{\mathrm{I}}^{\nu}, \bar{t}_{\mathrm{II}}^{\nu})}{f(\bar{s}^{\nu}, \bar{s}_{\mathrm{I}}^{\nu-1}) f(\bar{t}_{\mathrm{I}}^{\nu}, \bar{t}^{\nu-1})}$$

$$\times Z\big(\{\bar{s}_{\mathrm{II}}^k\}_1^{p-1}, \{\bar{s}^k\}_p^N \big| \{\bar{t}_{\mathrm{II}}^k\}_1^{p-1}; \{\bar{t}^k\}_p^N\big). \quad (4.15)$$

*In (4.15), for every fixed $p \in \{2,\ldots,N+1\}$ the sums are taken over partitions $\bar{t}^k \Rightarrow \{\bar{t}_{\mathrm{I}}^k, \bar{t}_{\mathrm{II}}^k\}$ with $k = 1,\ldots,p-1$ and $\bar{s}^k \Rightarrow \{\bar{s}_{\mathrm{I}}^k, \bar{s}_{\mathrm{II}}^k\}$ with $k = 2,\ldots,p-1$, such that $\#\bar{t}_{\mathrm{I}}^k = \#\bar{s}_{\mathrm{I}}^k = 1$ for $k = 2,\ldots,p-1$. The subset $\bar{s}_{\mathrm{I}}^1$ is a fixed Bethe parameter from the set $\bar{s}^1$. There is no sum over partitions of the set $\bar{s}^1$ in (4.15).*

The proof of this proposition coincides with the corresponding proof in [58].

**Corollary 4.8.** *The HC $Z(\bar{s}|\bar{t})$ satisfies the following recursion over the set $\bar{t}^N$:*

$$Z(\bar{s}|\bar{t}) = \sum_{p=1}^{N} \sum_{\substack{\text{part}(\bar{s}^p,\ldots,\bar{s}^N) \\ \text{part}(\bar{t}^p,\ldots,\bar{t}^{N-1})}} \frac{g^{(l)}(\bar{t}_{\mathrm{I}}^N, \bar{s}_{\mathrm{I}}^N) f(\bar{s}_{\mathrm{II}}^N, \bar{s}_{\mathrm{I}}^N) f(\bar{t}_{\mathrm{I}}^N, \bar{s}_{\mathrm{II}}^N)}{f(\bar{t}_{\mathrm{I}}^p, \bar{t}^{p-1})}$$

$$\times \prod_{\nu=p}^{N-1} \frac{g^{(l)}(\bar{s}_{\mathrm{I}}^{\nu+1}, \bar{s}_{\mathrm{I}}^{\nu}) g^{(r)}(\bar{t}_{\mathrm{I}}^{\nu+1}, \bar{t}_{\mathrm{I}}^{\nu}) f(\bar{s}_{\mathrm{II}}^{\nu}, \bar{s}_{\mathrm{I}}^{\nu}) f(\bar{t}_{\mathrm{I}}^{\nu}, \bar{t}_{\mathrm{II}}^{\nu})}{f(\bar{s}^{\nu+1}, \bar{s}_{\mathrm{I}}^{\nu}) f(\bar{t}_{\mathrm{I}}^{\nu+1}, \bar{t}^{\nu})}$$

$$\times Z\big(\{\bar{s}^k\}_1^{p-1}, \{\bar{s}_{\mathrm{II}}^k\}_p^N \big| \{\bar{t}^k\}_1^{p-1}; \{\bar{t}_{\mathrm{II}}^k\}_p^N\big). \quad (4.16)$$

*In (4.16), for every fixed $p \in \{1,\ldots,N\}$ the sums are taken over partitions $\bar{t}^k \Rightarrow \{\bar{t}_{\mathrm{I}}^k, \bar{t}_{\mathrm{II}}^k\}$ with $k = p,\ldots,N$ and $\bar{s}^k \Rightarrow \{\bar{s}_{\mathrm{I}}^k, \bar{s}_{\mathrm{II}}^k\}$ with $k = p,\ldots,N-1$, such that $\#\bar{t}_{\mathrm{I}}^k = \#\bar{s}_{\mathrm{I}}^k = 1$ for $k = p,\ldots,N-1$. The subset $\bar{t}_{\mathrm{I}}^N$ is a fixed Bethe parameter from the set $\bar{t}^N$. There is no sum over partitions for the set $\bar{t}^N$ in (4.16).*

This recursion follows from (4.15) and equation (4.14).

*Remark.* Similarly to the recursions for the Bethe vectors the sums over $p$ in (4.15), (4.16) break off, if HC $Z(\bar{s}|\bar{t})$ contains empty sets of the Bethe parameters with the colors $\{k_1,\ldots,k_\ell\}$, such that $k_1 < \cdots < k_\ell$. Namely, the sum over $p$ in (4.15) ends at $p = k_1$, while in (4.16) it begins at $p = k_\ell + 1$. These restrictions follow from the corresponding restrictions in the recursions for the Bethe vectors.

Using proposition 4.7 one can built the HC with $\#\bar{s}^1 = \#\bar{t}^1 = r_1$ in terms of the HC with $\#\bar{s}^1 = \#\bar{t}^1 = r_1 - 1$. Iterating the process, $Z(\bar{s}|\bar{t})$ with $\#\bar{s}^1 = \#\bar{t}^1 = r_1$ can be expressed in terms of $Z(\bar{s}|\bar{t})$ with $\#\bar{s}^1 = \#\bar{t}^1 = 0$. Moreover it is obvious, due to (4.6), that

$$Z^{(m)}(\emptyset, \{\bar{s}^k\}_2^N | \emptyset, \{\bar{t}^k\}_2^N) = Z^{(m-1)}(\{\bar{s}^k\}_2^N | \{\bar{t}^k\}_2^N), \quad (4.17)$$

where the superscript indicates for which algebra, $\mathcal{A}_m^q$ or $\mathcal{A}_{m-1}^q$, the HC is computed. Thus, equation (4.15) allows one to perform recursion over $m$ as well.

Similarly, corollary 4.8 allows one to find the HC with $\#\bar{s}^N = \#\bar{t}^N = r_N$ in terms of the HC with $\#\bar{s}^N = \#\bar{t}^N = r_N - 1$ and to perform another recursion over $m$. In both cases, the initial condition corresponds to the $\mathcal{A}_2^q$ case, where the HC is nothing but the Izergin–Korepin determinant [13, 60].

To conclude this section we describe the properties of HC in the poles.

**Proposition 4.9.** *The HC has poles at $s_j^\mu = t_j^\mu$, $\mu = 1, \ldots, N$, $j = 1, \ldots, r_\mu$. The residues in these poles are proportional to $Z(\bar{s} \setminus \{s_j^\mu\} | \bar{t} \setminus \{t_j^\mu\})$:*

$$Z(\bar{s}|\bar{t})\Big|_{s_j^\mu \to t_j^\mu} = g^{(l)}(t_j^\mu, s_j^\mu) \frac{f(\bar{t}_j^\mu, t_j^\mu) f(s_j^\mu, \bar{s}_j^\mu) \, Z(\bar{s} \setminus \{s_j^\mu\} | \bar{t} \setminus \{t_j^\mu\})}{f(\bar{t}^{\mu+1}, t_j^\mu) f(s_j^\mu, \bar{s}^{\mu-1})} + reg, \qquad (4.18)$$

*where $reg$ means regular terms.*

This property is in complete analogy with the Yangian case [59] and can be proved via induction and recursions (4.15), (4.16). In its turn, the residues of the HC play a crucial role in the proof of the Gaudin formula for the norm of on-shell Bethe vectors.

### 4.4 Norm of on-shell Bethe vectors and Gaudin matrix

The Gaudin matrix $G$ for $U_q(\widehat{\mathfrak{gl}}_m)$ based models is an $N \times N$ block-matrix. The sizes of the blocks $G^{(\mu,\nu)}$ are $r_\mu \times r_\nu$, where $r_\mu = \#\bar{t}^\mu$. To describe the entries $G_{jk}^{(\mu,\nu)}$ we introduce a function

$$\Phi_j^{(\mu)} = \alpha_\mu(t_j^\mu) \frac{f(\bar{t}_j^\mu, t_j^\mu)}{f(t_j^\mu, \bar{t}_j^\mu)} \frac{f(t_j^\mu, \bar{t}^{\mu-1})}{f(\bar{t}^{\mu+1}, t_j^\mu)}. \qquad (4.19)$$

It is easy to see that Bethe equations (3.19) can be written in terms of $\Phi_j^{(\mu)}$ as

$$\Phi_j^{(\nu)} = 1, \qquad j = 1, \ldots, r_\nu, \quad \nu = 1, \ldots, N. \qquad (4.20)$$

The entries of the Gaudin matrix are defined as

$$G_{jk}^{(\mu,\nu)} = -(q - q^{-1}) t_k^\nu \frac{\partial \log \Phi_j^{(\mu)}}{\partial t_k^\nu}. \qquad (4.21)$$

Explicitly, the diagonal blocks $G^{(\mu,\mu)}$ read

$$G_{jk}^{(\mu,\mu)} = \delta_{jk}\Big[ X_j^\mu - \sum_{p=1}^{r_\mu} \mathcal{K}(t_j^\mu, t_p^\mu) + \sum_{q=1}^{r_{\mu-1}} \mathcal{J}(t_j^\mu, t_q^{\mu-1}) + \sum_{r=1}^{r_{\mu+1}} \mathcal{J}(t_r^{\mu+1}, t_j^\mu) \Big] + \mathcal{K}(t_j^\mu, t_k^\mu), \quad (4.22)$$

while the off-diagonal blocks are given by

$$\begin{aligned} G_{jk}^{(\mu,\mu-1)} &= -\mathcal{J}(t_j^\mu, t_k^{\mu-1}), \qquad G_{jk}^{(\mu,\mu+1)} = -\mathcal{J}(t_k^{\mu+1}, t_j^\mu), \\ G_{jk}^{(\mu,\nu)} &= 0 \quad \text{if} \quad |\mu - \nu| > 1. \end{aligned} \qquad (4.23)$$

In (4.22) and (4.23), we have introduced the functions

$$X_j^\mu = -(q - q^{-1}) z \frac{d}{dz} \log \alpha_\mu(z)\Big|_{z=t_j^\mu}, \qquad (4.24)$$

$$\mathcal{K}(x,y) = \frac{(q + q^{-1})(q - q^{-1})^2 xy}{(qx - q^{-1}y)(q^{-1}x - qy)}, \quad \text{and} \quad \mathcal{J}(x,y) = \frac{(q - q^{-1})^2 xy}{(qx - q^{-1}y)(x - y)}. \qquad (4.25)$$

**Theorem 4.10.** *The square of the norm of the on-shell Bethe vector reads*

$$\mathbb{C}(\bar{t})\mathbb{B}(\bar{t}) = \prod_{k=1}^{N} \Big( f(\bar{t}^{k+1}, \bar{t}^k)^{-1} \prod_{\substack{p,q=1 \\ p \neq q}}^{r_k} f(t_p^k, t_q^k) \Big) \det G, \qquad (4.26)$$

*where the matrix $G$ is given by (4.21), or explicitly in (4.22) and (4.23).*

The proof of the similar theorem for the models described by the $Y(\mathfrak{gl}_m)$ and $Y(\mathfrak{gl}(m|n))$ $R$-matrices can be found in [59]. Despite the fact that in the case of $U_q(\widehat{\mathfrak{gl}}_m)$ algebra the proof is completely identical, we will briefly outline the main steps.

The main idea is to prove that the norm of on-shell Bethe vector satisfies several properties called *Korepin criteria*. Namely, let $\mathbf{F}^{(\mathbf{r})}(\bar{X};\bar{t})$ be a function depending on $\mathbf{r}$ variables $X_j^\mu$ and $\mathbf{r}$ variables $t_j^\mu$. It is assumed that this function satisfies Korepin criteria, if it possesses the following properties.

(i) The function $\mathbf{F}^{(\mathbf{r})}(\bar{X};\bar{t})$ is symmetric over the replacement of the pairs $(X_j^\mu, t_j^\mu) \leftrightarrow (X_k^\mu, t_k^\mu)$.

(ii) It is a linear function of each $X_j^\mu$.

(iii) $\mathbf{F}^{(1)}(X_1^1; t_1^1) = X_1^1$ for $\mathbf{r} = 1$.

(iv) The coefficient of $X_j^\mu$ is given by a function $\mathbf{F}^{(\mathbf{r}-1)}$ with modified parameters $X_k^\nu$

$$\frac{\partial \mathbf{F}^{(\mathbf{r})}(\bar{X};\bar{t})}{\partial X_j^\mu} = \mathbf{F}^{(\mathbf{r}-1)}(\{\bar{X}^{\text{mod}} \setminus X_j^{\text{mod};\mu}\}; \{\bar{t} \setminus t_j^\mu\}), \qquad (4.27)$$

where the original variables $X_k^\nu$ should be replaced by $X_k^{\text{mod};\nu}$:

$$
\begin{aligned}
X_k^{\text{mod};\mu} &= X_k^\mu - \mathcal{K}(t_j^\mu, t_k^\mu), \\
X_k^{\text{mod};\mu+1} &= X_k^{\mu+1} + \mathcal{J}(t_k^{\mu+1}, t_j^\mu), \\
X_k^{\text{mod};\mu-1} &= X_k^{\mu-1} + \mathcal{J}(t_j^\mu, t_k^{\mu-1}), \\
X_k^{\text{mod};\nu} &= X_k^\nu, \qquad |\nu - \mu| > 1.
\end{aligned}
\qquad (4.28)
$$

Here $\mathcal{K}(x,y)$ and $\mathcal{J}(x,y)$ are some two-variables functions. Their explicit forms are not essential.

(v) $\mathbf{F}^{(\mathbf{r})}(\bar{X};\bar{t}) = 0$, if all $X_j^\nu = 0$.

The properties (i)–(v) fix function $\mathbf{F}^{(\mathbf{r})}(\bar{X};\bar{t})$ uniquely (see [13, 59]). On the other hand, one can easily show that these properties are enjoyed by the determinant of the matrix $G$ given by equations (4.22), (4.23). Thus, $\mathbf{F}^{(\mathbf{r})}(\bar{X};\bar{t}) = \det G$.

The proof that the norm of the on-shell vector satisfies Korepin criteria is realized within the framework of the generalized model. In this model, Bethe parameters and logarithmic derivatives $X_j^\mu$ (4.24) are independent variables. Then properties (i)–(iii) are fairly obvious. Property (v) follows from the analysis of a special scalar product in which all $X_j^\mu = 0$. Finally, property (iv) is a consequence of the recursions of the highest coefficients with coinciding arguments (4.18). These recursions allow us to establish a recursion for the scalar product, which in turn implies property (iv) for the norm.

# 5 Proof of recursion for Bethe vectors

## 5.1 Proofs of proposition 4.1

One can prove proposition 4.1 via direct application of the nested algebraic Bethe ansatz. Let us briefly recall the basic notions of this method and introduce the necessary notation.

The nested algebraic Bethe ansatz relates Bethe vectors of $\mathscr{A}^q_m$ and $\mathscr{A}^q_{m-1}$ invariant systems. To distinguish objects associated to the $\mathscr{A}^q_{m-1}$ algebra from those from the $\mathscr{A}^q_m$ one, we use a special font for the former, keeping the usual style for the later. For example, we denote the basis vectors in $\mathbf{C}^m$ by $e_k$, where $(e_k)_j = \delta_{jk}$, and $j, k = 1, \ldots, m$, while the basis vectors in $\mathbf{C}^{m-1}$ are denoted by $\mathsf{e}_k$, where $(\mathsf{e}_k)_j = \delta_{jk}$, and $j, k = 2, \ldots, m$. Note that the enumeration of the basis vectors $\mathsf{e}_k$ starts at 2, not 1. We will use the same prescription for the other objects related to the $\mathscr{A}^q_{m-1}$ algebra and the $\mathbf{C}^{m-1}$ space.

We present the original monodromy matrix in the block form

$$T(u) = \begin{pmatrix} A(u) & B(u) \\ C(u) & D(u) \end{pmatrix}, \tag{5.1}$$

where $D(u)$ is a $(m-1) \times (m-1)$ matrix with elements $D_{i,j}(u)$, $i, j = 2, \ldots, m$.

Obviously, the elements $D_{i,j}(u)$ enjoy the commutation relations (2.4). Hence, the matrix $D(u)$ satisfies the $RTT$-relation

$$\mathsf{r}(u, v) \cdot (D(u) \otimes \mathbf{1}) \cdot (\mathbf{1} \otimes D(v)) = (\mathbf{1} \otimes D(v)) \cdot (D(u) \otimes \mathbf{1}) \cdot \mathsf{r}(u, v), \tag{5.2}$$

where $\mathsf{r}(u, v)$ is the $R$-matrix corresponding to the vector representation of the algebra $U_q(\widehat{\mathfrak{gl}}_{m-1})$

$$\begin{aligned}
\mathsf{r}(u, v) = {}& f(u, v) \sum_{2 \le i \le m} \mathsf{E}_{ii} \otimes \mathsf{E}_{ii} + \sum_{2 \le i < j \le m} \left( \mathsf{E}_{ii} \otimes \mathsf{E}_{jj} + \mathsf{E}_{jj} \otimes \mathsf{E}_{ii} \right) \\
& + \sum_{2 \le i < j \le m} g(u, v) \left( u \, \mathsf{E}_{ij} \otimes \mathsf{E}_{ji} + v \, \mathsf{E}_{ji} \otimes \mathsf{E}_{ij} \right).
\end{aligned} \tag{5.3}$$

In (5.3), $\mathsf{E}_{ij}$, $i, j = 2, \ldots, m$, are elementary units acting in $\mathbf{C}^{m-1}$, in accordance with the style convention described above.

Now we are in position to describe the main procedure of the nested algebraic Bethe ansatz. Let $\mathbb{B}(\bar{t}|T) = \mathbb{B}(\bar{t}^1, \ldots, \bar{t}^{m-1}|T)$ be a Bethe vector of the $U_q(\widehat{\mathfrak{gl}}_m)$ based monodromy matrix $T(u)$ such that $\#\bar{t}^\nu = r_\nu$. Let us introduce a Hilbert space

$$\mathscr{H}^{(r_1)} = \underbrace{\mathbf{C}^{m-1} \otimes \cdots \otimes \mathbf{C}^{m-1}}_{r_1}, \tag{5.4}$$

and an inhomogeneous monodromy matrix

$$T_{[r_1]}(u, \bar{t}^1) = \mathsf{r}_{0, r_1}(u, t^1_{r_1}) \ldots \mathsf{r}_{0,1}(u, t^1_1). \tag{5.5}$$

Remark that $T_{[r_1]}(u, \bar{t}^1)$ corresponds to a $U_q(\widehat{\mathfrak{gl}}_{m-1})$ model. Indeed, in (5.5), $\mathsf{r}_{0,k}(u, t^1_k)$ are the $R$-matrices (5.3) and they act in $\mathbf{C}^{m-1} \otimes \mathscr{H}^{(r_1)}$. The first subscript refers to an auxiliary space $\mathbf{C}^{m-1}$, while the second subscript refers to the $k$-th copy of $\mathbf{C}^{m-1}$ in the definition (5.4) of $\mathscr{H}^{(r_1)}$. It is clear that $T_{[r_1]}(u, \bar{t}^1)$ satisfies the $RTT$-relation (5.2).

Consider a monodromy matrix

$$\widetilde{T}_{[r_1]}(u, \bar{t}^1) = D(u) T_{[r_1]}(u, \bar{t}^1). \tag{5.6}$$

The entries of this matrix act in the space $\mathscr{H} \otimes \mathscr{H}^{(r_1)}$, where $\mathscr{H}$ is the space where the elements of the original monodromy matrix (5.1) act. It is clear that $\widetilde{T}_{[r_1]}(u, \bar{t}^1)$ satisfies the $RTT$ relation, because both $D(u)$ and $T_{[r_1]}(u, \bar{t}^1)$ satisfy this relation and their matrix elements act in the different quantum spaces (respectively in $\mathscr{H}$ and $\mathscr{H}^{(r_1)}$). The space of states of $\widetilde{T}_{[r_1]}$ has a pseudovacuum vector $|0\rangle \otimes \Omega_{r_1}$, where

$$\Omega_{r_1} = \underbrace{\mathsf{e}_2 \otimes \cdots \otimes \mathsf{e}_2}_{r_1} \in \left( \mathbf{C}^{m-1} \right)^{\otimes r_1}. \tag{5.7}$$

The subscript $r_1$ on $\Omega_{r_1}$ shows the number of copies of $\mathbf{C}^{m-1}$ in the space $\mathscr{H}^{(r_1)}$.

Let $\mathbb{B}(\bar{t}|\widetilde{T}_{[r_1]}) = \mathbb{B}(\bar{t}^2, \ldots, \bar{t}^{m-1}|\widetilde{T}_{[r_1]})$ be Bethe vectors of the monodromy matrix (5.6), and let $\widetilde{\alpha}_\nu^{(r_1-1)}(u)$ be the ratios of the vacuum eigenvalues of $\widetilde{T}_{[r_1-1]}(u)$. Then the Bethe vector $\mathbb{B}(\bar{t}|T)$ has the following presentation [29, 65]

$$
\mathbb{B}(\bar{t}|T) = \sum_{k_1,\ldots,k_{r_1}=2}^{m} \frac{T_{1,k_1}(t_1^1)\ldots T_{1,k_{r_1}}(t_{r_1}^1)}{\lambda_2(\bar{t}^1)f(\bar{t}^2,\bar{t}^1)} \left[\mathbb{B}(\bar{t}|\widetilde{T}_{[r_1]})\right]_{k_1,\ldots,k_{r_1}},
\tag{5.8}
$$

where $\left[\mathbb{B}(\bar{t}|\widetilde{T}_{[r_1]})\right]_{k_1,\ldots,k_{r_1}}$ are components of the vector $\mathbb{B}(\bar{t}|\widetilde{T}_{[r_1]})$ in the space $\mathscr{H}^{(r_1)}$.

Representation (5.8) allows us to obtain a recursion for the Bethe vector. This can be done in the framework of a composite model. Indeed, we have

$$
\widetilde{T}_{[r_1]}(u) = \widetilde{T}_{[r_1-1]}(u)\, r_{0,1}(u, t_1^1),
\tag{5.9}
$$

where

$$
\widetilde{T}_{[r_1-1]}(u) = D(u)T_{[r_1-1]}(u) = D(u)\, r_{0,r_1}(u, t_{r_1}^1)\ldots r_{0,2}(u, t_2^1).
\tag{5.10}
$$

We can associate the monodromy matrices $\widetilde{T}_{[r_1-1]}(u)$ and $r_{0,1}(u, t_1^1)$ respectively with $T^{(2)}(u)$ and $T^{(1)}(u)$ in (3.20). Then the partial Bethe vectors respectively are $\mathbb{B}(\bar{t}|\widetilde{T}_{[r_1-1]})$ and $\mathbb{B}(\bar{t}|r_{0,1})$. Using the coproduct formula (3.24) we obtain

$$
\begin{aligned}
\mathbb{B}(\bar{t}|T) = \sum_{k_1,\ldots,k_{r_1}=2}^{m} &\frac{T_{1,k_1}(t_1^1)\ldots T_{1,k_{r_1}}(t_{r_1}^1)}{\lambda_2(\bar{t}^1)f(\bar{t}^2,\bar{t}^1)} \\
\times &\sum_{\text{part}(\bar{t}^2,\ldots,\bar{t}^{m-1})} \frac{\prod_{\nu=2}^{m-1}\widetilde{\alpha}_\nu^{(r_1-1)}(\bar{t}_{\mathrm{I}}^\nu)f(\bar{t}_{\mathrm{II}}^\nu,\bar{t}_{\mathrm{I}}^\nu)}{\prod_{\nu=2}^{m-2}f(\bar{t}_{\mathrm{II}}^{\nu+1},\bar{t}_{\mathrm{I}}^\nu)}\left[\mathbb{B}(\bar{t}_{\mathrm{II}}|\widetilde{T}_{[r_1-1]})\right]_{k_2,\ldots,k_{r_1}}\left[\mathbb{B}(\bar{t}_{\mathrm{I}}|r_{0,1})\right]_{k_1}.
\end{aligned}
\tag{5.11}
$$

The sum is taken over partitions of the sets $\{\bar{t}^2,\ldots,\bar{t}^{m-1}\}$ as it is described in (3.24). The functions $\widetilde{\alpha}_\nu^{(r_1-1)}(u)$ are the ratios of the vacuum eigenvalues of $\widetilde{T}_{[r_1-1]}(u)$

$$
\widetilde{\alpha}_\nu^{(r_1-1)}(u) = \frac{\widetilde{\lambda}_\nu^{(r_1-1)}(u)}{\widetilde{\lambda}_{\nu+1}^{(r_1-1)}(u)},
\tag{5.12}
$$

where

$$
\left(\widetilde{T}_{[r_1-1]}(u)\right)_{\nu,\nu}|0\rangle \otimes \Omega_{r_1-1} = \widetilde{\lambda}_\nu^{(r_1-1)}(u)|0\rangle \otimes \Omega_{r_1-1},
\tag{5.13}
$$

and $\Omega_{r_1-1}$ is defined similarly to (5.7). It is convenient to divide the set $\bar{t}^1$ into two subsets $\bar{t}^1 = \bar{t}_{\mathrm{I}}^1 \cup \bar{t}_{\mathrm{II}}^1$, where $\bar{t}_{\mathrm{I}}^1$ consists of one element $t_1^1$, and $\bar{t}_{\mathrm{II}}^1 = \{t_2^1,\ldots,t_{r_1}^1\}$ is the complementary subset. Then it is easy to see from the definition (5.6) that

$$
\begin{aligned}
\widetilde{\lambda}_2^{(r_1-1)}(u) &= \lambda_2(u)f(u, \bar{t}_{\mathrm{II}}^1), \\
\widetilde{\lambda}_\nu^{(r_1-1)}(u) &= \lambda_\nu(u), \qquad \nu > 2,
\end{aligned}
\tag{5.14}
$$

and hence,

$$
\begin{aligned}
\widetilde{\alpha}_2^{(r_1-1)}(u) &= \alpha_2(u)f(u, \bar{t}_{\mathrm{II}}^1), \\
\widetilde{\alpha}_\nu^{(r_1-1)}(u) &= \alpha_\nu(u), \qquad \nu > 2.
\end{aligned}
\tag{5.15}
$$

Due to (5.8) we see that

$$
\sum_{k_2,\ldots,k_{r_1}=2}^{m} \frac{T_{1,k_2}(t_2^1)\ldots T_{1,k_{r_1}}(t_{r_1}^1)}{\lambda_2(\bar{t}_{\mathrm{II}}^1)f(\bar{t}_{\mathrm{II}}^2,\bar{t}_{\mathrm{II}}^1)}\left[\mathbb{B}(\bar{t}_{\mathrm{II}}|\widetilde{T}_{[r_1-1]})\right]_{k_2,\ldots,k_{r_1}} = \mathbb{B}(\bar{t}_{\mathrm{II}}|T).
\tag{5.16}
$$

Substituting this into (5.11) we find

$$\mathbb{B}(\bar{t}|T) = \sum_{\text{part}(\bar{t}^2,\dots,\bar{t}^{m-1})} \sum_{k=2}^{m} \frac{T_{1,k}(t_{\text{I}}^1)}{\lambda_2(t_{\text{I}}^1)} \mathbb{B}(\bar{t}_{\text{II}}|T) \frac{\prod_{\nu=2}^{m-1} \alpha_\nu(\bar{t}_{\text{I}}^\nu) f(\bar{t}_{\text{II}}^\nu, \bar{t}_{\text{I}}^\nu)}{\prod_{\nu=2}^{m-2} f(\bar{t}_{\text{II}}^{\nu+1}, \bar{t}_{\text{I}}^\nu)} \frac{[\mathbb{B}(\bar{t}_{\text{I}}|r_{0,1})]_k}{f(\bar{t}^2, \bar{t}_{\text{I}}^1)}. \tag{5.17}$$

The components of the vector $\mathbb{B}(\bar{t}_{\text{I}}|r_{0,1})$ are computed in appendix A (see (A.4)). It follows from these formulas that the $k$-th component of this vector corresponds to the partitions for which the subsets $\bar{t}_{\text{I}}^k, \dots, \bar{t}_{\text{I}}^{m-1}$ are empty, while the subsets $\bar{t}_{\text{I}}^\nu$ with $2 \leq \nu < k$ consist of one element. This gives us

$$\mathbb{B}(\bar{t}|T) = \sum_{\text{part}(\bar{t}^2,\dots,\bar{t}^{m-1})} \sum_{k=2}^{m} \frac{T_{1,k}(t_{\text{I}}^1)}{\lambda_2(t_{\text{I}}^1)} \mathbb{B}(\{\bar{t}_{\text{II}}^\nu\}_1^{k-1}; \{\bar{t}^\nu\}_k^{m-1}|T) \frac{\prod_{\nu=2}^{k-1} \alpha_\nu(\bar{t}_{\text{I}}^\nu) g^{(l)}(\bar{t}_{\text{I}}^\nu, \bar{t}_{\text{I}}^{\nu-1}) f(\bar{t}_{\text{II}}^\nu, \bar{t}_{\text{I}}^\nu)}{\prod_{\nu=1}^{k-1} f(\bar{t}^{\nu+1}, \bar{t}_{\text{I}}^\nu)}. \tag{5.18}$$

Recall that here by definition the subsets $\bar{t}_{\text{I}}^1$ and $\bar{t}_{\text{II}}^1$ are fixed: $\bar{t}_{\text{I}}^1 \equiv t_1^1$ and $\bar{t}_{\text{II}}^1 \equiv \bar{t}_{\text{I}}^1 = \bar{t}^1 \setminus t_1^1$. Then, replacing $\bar{t}^1 \to \{z, \bar{t}^1\}$ and setting $\bar{t}_{\text{I}}^1 = z$ we arrive at (4.1).

## 5.2 Proofs of proposition 4.2

Let us derive now the recursion (4.3) starting with (4.1) and using the morphism (3.9). The proof mimics the one done in [58], and we just point out the differences. Since the mapping (3.9) relates two different quantum algebras $\mathscr{A}_m^q$ and $\mathscr{A}_m^{q^{-1}}$, we use here an additional subscript for the different rational functions, to denote the value of the deformation parameter. For instance

$$f_q(u,v) = \frac{qu - q^{-1}v}{u - v}, \quad \text{and} \quad g_q(u,v) = \frac{q - q^{-1}}{u - v}, \tag{5.19}$$

while

$$f_{q^{-1}}(u,v) = \frac{q^{-1}u - qv}{u - v}, \quad \text{and} \quad g_{q^{-1}}(u,v) = \frac{q^{-1} - q}{u - v}. \tag{5.20}$$

It is easy to see that

$$g_{q^{-1}}^{(r)}(u,v) = g_q^{(l)}(v,u) \quad \text{and} \quad f_{q^{-1}}(u,v) = f_q(v,u). \tag{5.21}$$

We act with $\varphi$ onto (4.1) using (3.9)–(3.11). It implies in particular

$$\varphi\left(\mathbb{B}_q(\{\bar{t}^1\}; \{\bar{t}_{\text{II}}^k\}_2^{j-1}; \{\bar{t}^k\}_j^N) \prod_{\nu=2}^{j-1} \alpha_\nu(\bar{t}_{\text{I}}^\nu)\right) = \frac{\mathbb{B}_{q^{-1}}(\{\bar{t}^k\}_N^j; \{\bar{t}_{\text{II}}^k\}_{j-1}^2; \bar{t}^1)}{\prod_{k=1}^N \widetilde{\alpha}_{N+1-k}(\bar{t}^k)}. \tag{5.22}$$

Remark that the functions $\alpha_\nu$ play a non-trivial role in the game. Then, the action of the morphism $\varphi$ onto (4.1) gives

$$\mathbb{B}_{q^{-1}}(\{\bar{t}^k\}_N^2; \{z, \bar{t}^1\}) = \sum_{j=2}^{N+1} \frac{\widetilde{T}_{N+2-j, N+1}(z)}{\widetilde{\lambda}_{N+1}(z)} \sum_{\text{part}(\bar{t}^2,\dots,\bar{t}^{j-1})} \mathbb{B}_{q^{-1}}(\{\bar{t}^k\}_N^j; \{\bar{t}_{\text{II}}^k\}_{j-1}^2; \bar{t}^1)$$

$$\times \frac{\prod_{\nu=2}^{j-1} g_q^{(l)}(\bar{t}_{\text{I}}^\nu, \bar{t}_{\text{I}}^{\nu-1}) f_q(\bar{t}_{\text{II}}^\nu, \bar{t}_{\text{I}}^\nu)}{\prod_{\nu=1}^{j-1} f_q(\bar{t}^{\nu+1}, \bar{t}_{\text{I}}^\nu)}. \tag{5.23}$$

Using the relations (5.21), relabeling the sets of the Bethe parameters $\bar{t}^k \to \bar{t}^{N+1-k}$, changing indices $j \to N+2-j$, $\nu \to N+1-\nu$ and replacing $q^{-1} \to q$ (which means going from $\mathscr{A}_m^{q^{-1}}$ to $\mathscr{A}_m^q$) we get (4.3). $\qquad \square$

### 5.3 Proofs of corollary 4.3

The proof for corollary 4.3 follows the same steps as in section 5.2, but using the antimorphism $\Psi$ instead of the morphism $\varphi$. Thus, we just sketch the proof.

One starts with relation (4.1) and applies $\Psi$, to get in $\mathscr{A}_m^{q^{-1}}$:

$$
\mathbb{C}_{q^{-1}}\Big(\Big\{\frac{1}{z},\frac{1}{\bar{t}^1}\Big\};\Big\{\frac{1}{\bar{t}^k}\Big\}_2^N\Big) = \sum_{j=2}^{N+1} \sum_{\text{part}(\bar{t}^2,\dots,\bar{t}^{j-1})} \mathbb{C}_{q^{-1}}\Big(\Big\{\frac{1}{\bar{t}^1}\Big\};\Big\{\frac{1}{\bar{t}_{\text{II}}^k}\Big\}_2^{j-1};\Big\{\frac{1}{\bar{t}^k}\Big\}_j^N\Big) \frac{\widetilde{T}_{j,1}(\frac{1}{z})}{\widetilde{\lambda}_2(\frac{1}{z})}
$$
$$
\times \frac{\prod_{\nu=2}^{j-1}\widetilde{\alpha}_\nu(\frac{1}{\bar{t}_1^\nu})\, g_q^{(l)}(\bar{t}_1^\nu,\bar{t}_1^{\nu-1}) f_q(\bar{t}_{\text{II}}^\nu,\bar{t}_1^\nu)}{\prod_{\nu=1}^{j-1} f_q(\bar{t}^{\nu+1},\bar{t}_1^\nu)}. \quad (5.24)
$$

Now, renaming the parameters $t_k^\nu \to \frac{1}{t_k^\nu}$, $z \to \frac{1}{z}$ and using the relations

$$
g_q^{(r)}\Big(\frac{1}{x},\frac{1}{y}\Big) = g_{q^{-1}}^{(l)}(x,y) \quad \text{and} \quad f_q\Big(\frac{1}{x},\frac{1}{y}\Big) = f_{q^{-1}}(x,y) \quad (5.25)
$$

we obtain

$$
\mathbb{C}_{q^{-1}}(\{z,\bar{t}^1\};\{\bar{t}^k\}_2^N) = \sum_{j=2}^{N+1} \sum_{\text{part}(\bar{t}^2,\dots,\bar{t}^{j-1})} \mathbb{C}_{q^{-1}}(\{\bar{t}^1\};\{\bar{t}_{\text{II}}^k\}_2^{j-1};\{\bar{t}^k\}_j^N) \frac{\widetilde{T}_{j,1}(z)}{\widetilde{\lambda}_2(z)}
$$
$$
\times \frac{\prod_{\nu=2}^{j-1}\widetilde{\alpha}_\nu(\bar{t}_1^\nu)\, g_{q^{-1}}^{(r)}(\bar{t}_1^\nu,\bar{t}_1^{\nu-1}) f_{q^{-1}}(\bar{t}_{\text{II}}^\nu,\bar{t}_1^\nu)}{\prod_{\nu=1}^{j-1} f_{q^{-1}}(\bar{t}^{\nu+1},\bar{t}_1^\nu)}. \quad (5.26)
$$

It remains to change $q^{-1} \to q$ to get relation (4.4). Similar considerations lead to (4.5). $\qquad\square$

## 6 Proof of proposition 4.5

In this section we provide an explicit representation of the rational coefficients $W_{\text{part}}$ (4.8) in terms of the HC. For this we consider the original monodromy matrix $T(u)$ as a monodromy matrix of a composite model (3.20). Then we should use the representation (3.24) for the Bethe vector $\mathbb{B}(\bar{t})$ and the representation (3.25) for the dual vector $\mathbb{C}(\bar{s})$. As a consequence, the scalar product $S(\bar{s}|\bar{t}) = \mathbb{C}(\bar{s})\mathbb{B}(\bar{t})$ takes the form

$$
S(\bar{s}|\bar{t}) = \sum \frac{\prod_{\nu=1}^{N} \alpha_\nu^{(1)}(\bar{s}_{\text{ii}}^\nu)\alpha_\nu^{(2)}(\bar{t}_{\text{i}}^\nu) f(\bar{s}_{\text{i}}^\nu,\bar{s}_{\text{ii}}^\nu) f(\bar{t}_{\text{ii}}^\nu,\bar{t}_{\text{i}}^\nu)}{\prod_{\nu=1}^{N-1} f(\bar{s}_{\text{i}}^{\nu+1},\bar{s}_{\text{ii}}^\nu) f(\bar{t}_{\text{ii}}^{\nu+1},\bar{t}_{\text{i}}^\nu)} S^{(1)}(\bar{s}_{\text{i}}|\bar{t}_{\text{i}}) S^{(2)}(\bar{s}_{\text{ii}}|\bar{t}_{\text{ii}}), \quad (6.1)
$$

where

$$
S^{(1)}(\bar{s}_{\text{i}}|\bar{t}_{\text{i}}) = \mathbb{C}(\bar{s}_{\text{i}}|T^{(1)})\mathbb{B}(\bar{t}_{\text{i}}|T^{(1)}), \qquad S^{(2)}(\bar{s}_{\text{ii}}|\bar{t}_{\text{ii}}) = \mathbb{C}(\bar{s}_{\text{ii}}|T^{(2)})\mathbb{B}(\bar{t}_{\text{ii}}|T^{(2)}). \quad (6.2)
$$

Note that in this formula $\#\bar{s}_{\text{i}}^\nu = \#\bar{t}_{\text{i}}^\nu$, (and hence, $\#\bar{s}_{\text{ii}}^\nu = \#\bar{t}_{\text{ii}}^\nu$), otherwise the scalar products $S^{(1)}$ and $S^{(2)}$ vanish. Let $\#\bar{s}_{\text{i}}^\nu = \#\bar{t}_{\text{i}}^\nu = k_\nu'$, where $k_\nu' = 0,1,\dots,r_\nu$. Then $\#\bar{s}_{\text{ii}}^\nu = \#\bar{t}_{\text{ii}}^\nu = r_\nu - k_\nu'$.

Now let us turn to equation (4.8). Our goal is to express the rational coefficients $W_{\text{part}}$ in terms of the HC. For this we use the fact that $W_{\text{part}}$ are model independent. Therefore, we can find them in some special model whose monodromy matrix satisfies the $RTT$-relation.

Let us fix some partitions of the Bethe parameters in (4.8): $\bar{s}^\nu \Rightarrow \{\bar{s}_{\text{i}}^\nu,\bar{s}_{\text{II}}^\nu\}$ and $\bar{t}^\nu \Rightarrow \{\bar{t}_{\text{i}}^\nu,\bar{t}_{\text{II}}^\nu\}$ such that $\#\bar{s}_{\text{i}}^\nu = \#\bar{t}_{\text{i}}^\nu = k_\nu$, for some $k_\nu = 0,1,\dots,r_\nu$. Hence, $\#\bar{s}_{\text{II}}^\nu = \#\bar{t}_{\text{II}}^\nu = r_\nu - k_\nu$. Consider a concrete model, in which

$$
\begin{aligned}
\alpha_\nu^{(1)}(z) &= 0, \quad \text{if} \quad z \in \bar{s}_{\text{II}}^\nu, \\
\alpha_\nu^{(2)}(z) &= 0, \quad \text{if} \quad z \in \bar{t}_{\text{I}}^\nu.
\end{aligned} \quad (6.3)
$$

Due to (3.23) these conditions imply

$$\alpha_\nu(z) = 0, \quad \text{if} \quad z \in \bar{s}_{\mathrm{II}}^\nu \cup \bar{t}_{\mathrm{I}}^\nu. \tag{6.4}$$

Then the scalar product is proportional to the coefficient $W_{\mathrm{part}}(\bar{s}_{\mathrm{I}}, \bar{s}_{\mathrm{II}} | \bar{t}_{\mathrm{I}}, \bar{t}_{\mathrm{II}})$, because all other terms in the sum over partitions (4.8) vanish due to the condition (6.4). Thus,

$$S(\bar{s}|\bar{t}) = W_{\mathrm{part}}(\bar{s}_{\mathrm{I}}, \bar{s}_{\mathrm{II}} | \bar{t}_{\mathrm{I}}, \bar{t}_{\mathrm{II}}) \prod_{k=1}^{N} \alpha_k(\bar{s}_{\mathrm{I}}^k) \alpha_k(\bar{t}_{\mathrm{II}}^k). \tag{6.5}$$

On the other hand, (6.3) implies that a non-zero contribution in (6.1) occurs if and only if $\bar{s}_{\mathrm{ii}}^\nu \subset \bar{s}_{\mathrm{I}}^\nu$ and $\bar{t}_{\mathrm{i}}^\nu \subset \bar{t}_{\mathrm{II}}^\nu$. Hence, $r_\nu - k_\nu' \le k_\nu$ and $k_\nu' \le r_\nu - k_\nu$. But this is possible if and only if $k_\nu' + k_\nu = r_\nu$. Thus, $\bar{s}_{\mathrm{ii}}^\nu = \bar{s}_{\mathrm{I}}^\nu$ and $\bar{t}_{\mathrm{i}}^\nu = \bar{t}_{\mathrm{II}}^\nu$. Then, for the complementary subsets we obtain $\bar{s}_{\mathrm{i}}^\nu = \bar{s}_{\mathrm{II}}^\nu$ and $\bar{t}_{\mathrm{ii}}^\nu = \bar{t}_{\mathrm{I}}^\nu$. Thus, we arrive at

$$S(\bar{s}|\bar{t}) = \frac{\prod_{\nu=1}^{N} \alpha_\nu^{(1)}(\bar{s}_{\mathrm{I}}^\nu) \alpha_\nu^{(2)}(\bar{t}_{\mathrm{II}}^\nu) f(\bar{s}_{\mathrm{II}}^\nu, \bar{s}_{\mathrm{I}}^\nu) f(\bar{t}_{\mathrm{I}}^\nu, \bar{t}_{\mathrm{II}}^\nu)}{\prod_{\nu=1}^{N-1} f(\bar{s}_{\mathrm{II}}^{\nu+1}, \bar{s}_{\mathrm{I}}^\nu) f(\bar{t}_{\mathrm{I}}^{\nu+1}, \bar{t}_{\mathrm{II}}^\nu)} S^{(1)}(\bar{s}_{\mathrm{II}}|\bar{t}_{\mathrm{II}}) S^{(2)}(\bar{s}_{\mathrm{I}}|\bar{t}_{\mathrm{I}}). \tag{6.6}$$

It is easy to see that calculating the scalar product $S^{(1)}(\bar{s}_{\mathrm{II}}|\bar{t}_{\mathrm{II}})$ we should take only the term corresponding to the conjugated HC. Indeed, all other terms are proportional to $\alpha_\nu^{(1)}(z)$ with $z \in \bar{s}_{\mathrm{II}}^\nu$, therefore, they vanish. Hence

$$S^{(1)}(\bar{s}_{\mathrm{II}}|\bar{t}_{\mathrm{II}}) = \prod_{\nu=1}^{N} \alpha_\nu^{(1)}(\bar{t}_{\mathrm{II}}^\nu) \cdot \overline{Z}(\bar{s}_{\mathrm{II}}|\bar{t}_{\mathrm{II}}). \tag{6.7}$$

Similarly, calculating the scalar product $S^{(2)}(\bar{s}_{\mathrm{I}}|\bar{t}_{\mathrm{I}})$ we should take only the term corresponding to the HC:

$$S^{(2)}(\bar{s}_{\mathrm{I}}|\bar{t}_{\mathrm{I}}) = \prod_{\nu=1}^{N} \alpha_\nu^{(2)}(\bar{s}_{\mathrm{I}}^\nu) \cdot Z(\bar{s}_{\mathrm{I}}|\bar{t}_{\mathrm{I}}). \tag{6.8}$$

Substituting this into (6.6) and using (3.23), (6.5) we arrive at (4.11).

The reader can easily convince himself that the above proof coincides with the one given in [58] for the $Y(\mathfrak{gl}(m|n))$ based models.

As already mentioned, the proofs for the results presented in section 4.2 and 4.4 are also similar to those of the $Y(\mathfrak{gl}(m|n))$ based models and given in [58, 59], thus we don't repeat them here. In the following section we deal with the proof for section 4.3, focusing on the parts that truly differ from the Yangian case.

## 7 Symmetry of the highest coefficient

To prove (4.12), we consider the sum formula (4.8)

$$S_q(\vec{s}|\vec{t}) = \sum W_{\mathrm{part}}^q(\vec{s}_{\mathrm{I}}, \vec{s}_{\mathrm{II}} | \vec{t}_{\mathrm{I}}, \vec{t}_{\mathrm{II}}) \prod_{k=1}^{N} \alpha_k(\bar{s}_{\mathrm{I}}^k) \alpha_k(\bar{t}_{\mathrm{II}}^k), \tag{7.1}$$

where we have stressed the ordering (3.12) of the Bethe parameters and put a label $q$ to distinguish scalar product for the algebra $\mathscr{A}_m^q$ from $\mathscr{A}_m^{q^{-1}}$. Let us act with the morphism $\varphi$ (3.9) on this scalar product. This can be done in two ways. First, using (3.11) and (3.18) we

obtain

$$
\varphi\Big(S_q(\vec{s}|\vec{t})\Big) = \varphi\Big(\mathbb{C}_q(\vec{s})\mathbb{B}_q(\vec{t})\Big) = \frac{\mathbb{C}_{q^{-1}}(\overleftarrow{s})\mathbb{B}_{q^{-1}}(\overleftarrow{t})}{\prod_{k=1}^{N}\widetilde{\alpha}_{N+1-k}(\bar{s}^k)\widetilde{\alpha}_{N+1-k}(\bar{t}^k)}
$$
$$
= \frac{S_{q^{-1}}(\overleftarrow{s}|\overleftarrow{t})}{\prod_{k=1}^{N}\widetilde{\alpha}_{N+1-k}(\bar{s}^k)\widetilde{\alpha}_{N+1-k}(\bar{t}^k)}. \tag{7.2}
$$

The scalar product $S_{q^{-1}}(\overleftarrow{s}|\overleftarrow{t})$ has the standard representation (4.8). Thus, we find

$$
\varphi\Big(S_q(\vec{s}|\vec{t})\Big) = \sum_{\text{part}} \frac{W_{\text{part}}^{q^{-1}}(\overleftarrow{s}_{\text{I}},\overleftarrow{s}_{\text{II}}|\overleftarrow{t}_{\text{I}},\overleftarrow{t}_{\text{II}})}{\prod_{k=1}^{N}\widetilde{\alpha}_{N+1-k}(\bar{s}^k)\widetilde{\alpha}_{N+1-k}(\bar{t}^k)} \prod_{k=1}^{N} \widetilde{\alpha}_k(\bar{s}_{\text{I}}^{N-k+1})\widetilde{\alpha}_k(\bar{t}_{\text{II}}^{N-k+1}). \tag{7.3}
$$

On the other hand, acting with $\varphi$ directly on the sum formula (7.1) we have

$$
\varphi\Big(S_q(\vec{s}|\vec{t})\Big) = \sum_{\text{part}} W_{\text{part}}^{q}(\vec{s}_{\text{I}},\vec{s}_{\text{II}}|\vec{t}_{\text{I}},\vec{t}_{\text{II}}) \prod_{k=1}^{N}\Big(\widetilde{\alpha}_{N+1-k}(\bar{s}_{\text{I}}^k)\widetilde{\alpha}_{N+1-k}(\bar{t}_{\text{II}}^k)\Big)^{-1}. \tag{7.4}
$$

Comparing (7.3) and (7.4) we arrive at

$$
\sum_{\text{part}} W_{\text{part}}^{q^{-1}}(\overleftarrow{s}_{\text{I}},\overleftarrow{s}_{\text{II}}|\overleftarrow{t}_{\text{I}},\overleftarrow{t}_{\text{II}}) \prod_{k=1}^{N} \widetilde{\alpha}_{N+1-k}(\bar{s}_{\text{I}}^k)\widetilde{\alpha}_{N+1-k}(\bar{t}_{\text{II}}^k)
$$
$$
= \sum_{\text{part}} W_{\text{part}}^{q}(\vec{s}_{\text{I}},\vec{s}_{\text{II}}|\vec{t}_{\text{I}},\vec{t}_{\text{II}}) \prod_{k=1}^{N} \widetilde{\alpha}_{N+1-k}(\bar{s}_{\text{II}}^k)\widetilde{\alpha}_{N+1-k}(\bar{t}_{\text{I}}^k). \tag{7.5}
$$

Since $\alpha_i$ are free functional parameters, the coefficients of the same products of $\widetilde{\alpha}_i$ must be equal. Hence,

$$
W_{\text{part}}^{q}(\vec{s}_{\text{I}},\vec{s}_{\text{II}}|\vec{t}_{\text{I}},\vec{t}_{\text{II}}) = W_{\text{part}}^{q^{-1}}(\overleftarrow{s}_{\text{II}},\overleftarrow{s}_{\text{I}}|\overleftarrow{t}_{\text{II}},\overleftarrow{t}_{\text{I}}), \tag{7.6}
$$

for arbitrary partitions of the sets $\bar{s}$ and $\bar{t}$. In particular, setting $\bar{s}_{\text{II}} = \bar{t}_{\text{II}} = \emptyset$ we obtain (4.12).

To prove (4.13), we start again with the sum formula (4.8) and use the antimorphism $\Psi$:

$$
\Psi(S_q(\bar{s}|\bar{t})) = \mathbb{C}_{q^{-1}}(\bar{t}^{-1})\mathbb{B}_{q^{-1}}(\bar{s}^{-1}) = S_{q^{-1}}(\bar{t}^{-1}|\bar{s}^{-1}). \tag{7.7}
$$

The lhs of (7.7) can be computed from the relation (4.8):

$$
\Psi(S_q(\bar{s}|\bar{t})) = \sum W_{\text{part}}^{q}(\bar{s}_{\text{I}},\bar{s}_{\text{II}}|\bar{t}_{\text{I}},\bar{t}_{\text{II}}) \prod_{k=1}^{N} \widetilde{\alpha}_k\Big(\frac{1}{\bar{s}_{\text{I}}^k}\Big)\widetilde{\alpha}_k\Big(\frac{1}{\bar{t}_{\text{II}}^k}\Big). \tag{7.8}
$$

The rhs of (7.7) is computed directly from (4.8) written for $\mathscr{A}_m^{q^{-1}}$:

$$
S_{q^{-1}}(\bar{t}^{-1}|\bar{s}^{-1}) = \sum W_{\text{part}}^{q^{-1}}(\bar{t}_{\text{I}}^{-1},\bar{t}_{\text{II}}^{-1}|\bar{s}_{\text{I}}^{-1},\bar{s}_{\text{II}}^{-1}) \prod_{k=1}^{N} \widetilde{\alpha}_k\Big(\frac{1}{\bar{t}_{\text{I}}^k}\Big)\widetilde{\alpha}_k\Big(\frac{1}{\bar{s}_{\text{II}}^k}\Big). \tag{7.9}
$$

Since $\alpha_i$ are free functional parameters, the comparison of these two equalities leads to

$$
W_{\text{part}}^{q}(\bar{s}_{\text{I}},\bar{s}_{\text{II}}|\bar{t}_{\text{I}},\bar{t}_{\text{II}}) = W_{\text{part}}^{q^{-1}}(\bar{t}_{\text{II}}^{-1},\bar{t}_{\text{I}}^{-1}|\bar{s}_{\text{II}}^{-1},\bar{s}_{\text{I}}^{-1}). \tag{7.10}
$$

Setting $\bar{s}_{\text{I}} = \bar{t}_{\text{I}} = \emptyset$, we get (4.13).

Combining (4.12) and (4.13), we get (4.14).

Applying the property (4.14) to (4.15), one obtains a new recursion written for the parameters $\bar{t}^{-1}$ and $\bar{s}^{-1}$. Using the relations

$$
g^{(l)}\Big(\frac{1}{x},\frac{1}{y}\Big) = g^{(l)}(y,x) \quad \text{and} \quad f\Big(\frac{1}{x},\frac{1}{y}\Big) = f(y,x)
$$

together with the replacement $\bar{t}^{-1} \to \bar{t}$ and $\bar{s}^{-1} \to \bar{s}$, we get the recursion (4.16) for the highest coefficient.

## Conclusion

In this paper, we have shown how the results obtained for the scalar products and the norm of Bethe vectors for $Y(\mathfrak{gl}(m))$ based models can be generalized to the case of $U_q(\widehat{\mathfrak{gl}}_m)$ based models. In this way, we have obtained recursion formulas for the Bethe vectors of these models, as well as a sum formula for their scalar products. We have obtained different recursions for the highest coefficients, which characterize the sum formula. When the Bethe vectors are on-shell, we have also shown that their norm takes the form of a Gaudin determinant.

Comparing these results with the ones obtained for the case of $Y(\mathfrak{gl}(m))$, one can see that for the most of them the generalization is quite straightforward. The only minor difference is that in the Yangian case the highest coefficient of the scalar product coincides with its conjugated, while for the $\mathscr{A}_m^q$ algebra they are related by the transformations (4.12), (4.13). This difference was already pointed out in [49] for the particular case of the $U_q(\widehat{\mathfrak{gl}}_3)$ based models.

The sum formula itself is rather bulky, however, we recall that it is obtained for the most general case of the Bethe vectors scalar product. This formula can be used as a starting point for calculating form factors of the monodromy matrix entries. In this case we deal with scalar products involving on-shell Bethe vectors. Then, the free functional parameters $\alpha_k(u)$ disappear from the sum formula due to Bethe equations, and we obtain a possibility for additional re-summation. This re-summation might lead to compact determinant representations for form factors (see e.g. [50] for the $\mathscr{A}_3^q$ case), like in the case of the norm of on-shell Bethe vector.

One more possible simplification of the sum formula is related to consideration of specific models, in which the free functional parameters $\alpha_k(u)$ are fixed. For instance, for the spin chain based on $U_q(\widehat{\mathfrak{gl}}_m)$ fundamental representations, $\alpha_1(u)$ is a rational function, while $\alpha_k(u) = 1$ for $k > 1$. Thus, in this case most of these functional parameters also disappear from the sum formula, which gives a chance for its simplification.

These two possibilities of further development certainly are worthy of attention. Finally, we wish to note that it seems to us rather obvious that the results presented here can also be readily generalized to the case of models based on $U_q(\widehat{\mathfrak{gl}}(m|n))$. We plan to come back on this generalization in a further publication.

## Acknowledgements

The work of A.L. has been funded by Russian Academic Excellence Project 5-100, by Young Russian Mathematics award and by joint NASU-CNRS project F14-2017. The work of S.P. was supported in part by the RFBR grant 16-01-00562-a.

## A The simplest $U_q(\widehat{\mathfrak{gl}}_m)$ Bethe vectors

In this section we construct Bethe vectors for a very specific case of the $\mathscr{A}_m^q$ monodromy matrix $T(u) = R(u, \xi)$, where $R(u, \xi)$ is given by (2.1) and $\xi$ is a complex number. In other words, we consider spin chain with only one site which carries a fundamental representation of $\mathscr{A}_m^q$. The Bethe vector construction procedure is still based on the embedding (5.1) of $\mathscr{A}_{m-1}^q$ into $\mathscr{A}_m^q$. In this appendix, to distinguish Bethe vectors corresponding to the $R$-matrices (2.1) and (5.3) we respectively equip them with superscripts $(m)$ or $(m-1)$.

This case has many peculiarities which allow a simple and explicit calculation of Bethe vectors. First of all, the space of states is $\mathscr{H} = \mathbf{C}^m$ with the pseudovacuum $|0\rangle = e_1$. As usual, the Bethe vectors depend on $N = m - 1$ sets of variables $\bar{t}^\nu$. However, due to the nilpotency

of the creation operators[6] each set consists at most of one element. Furthermore, $D_{i,i}|0\rangle = |0\rangle$ for all $i = 2, \ldots, m$. Therefore, in the framework of the algebraic Bethe ansatz, the matrix $D$ is equivalent to the identity matrix. Hence, we can omit this matrix in the definition (5.6).

**Proposition A.1.** *The monodromy matrix* $T(u) = R(u, \xi)$ *has* $m - 1$ *Bethe vectors of the form*

$$\mathbb{B}^{(m)}(\{t^\nu\}_1^{k-1}, \{\emptyset\}_k^{m-1}) = \left(\prod_{\nu=2}^{k-1} \frac{g^{(l)}(t^\nu, t^{\nu-1})}{f(t^\nu, t^{\nu-1})}\right) g^{(l)}(t^1, \xi) e_k, \qquad k = 2, \ldots, m. \tag{A.1}$$

*One additional Bethe vector coincides with the pseudovacuum* $e_1$.

*Proof.* One can easily prove (A.1) via induction over $m$. Indeed, for $m = 2$ we have only two Bethe vectors: the pseudovacuum $e_1 = \binom{1}{0} \in \mathbb{C}^2$ and

$$\mathbb{B}^{(2)}(t^1) = T_{12}(t^1) e_1 = g^{(l)}(t^1, \xi) E_{21} e_1 = g^{(l)}(t^1, \xi) e_2 = g^{(l)}(t^1, \xi) \binom{0}{1}. \tag{A.2}$$

Assume that (A.1) holds for $m - 1$. One of the $U_q(\widehat{\mathfrak{gl}}_m)$ Bethe vectors still coincides with the pseudovacuum vector $\mathbb{B}^{(m)}(\emptyset) = e_1$. The other Bethe vectors can be constructed via (5.8), where one should set $\lambda_2(u) = 1$:

$$\mathbb{B}^{(m)}(t^1, \ldots, t^{m-1}) = \sum_{k=2}^{m} T_{1,k}(t^1) e_1 \frac{\left[\mathbb{B}^{(m-1)}(t^2, \ldots, t^{m-1})\right]_k}{f(t^2, t^1)}. \tag{A.3}$$

Here $\left[\mathbb{B}^{(m-1)}(t^2, \ldots, t^{m-1})\right]_k$ is the $k$-th component of the Bethe vector $\mathbb{B}^{(m-1)}(\bar{t})$ of the monodromy matrix $r(u, t^1)$ (5.3). Due to the induction assumption we have

$$\left[\mathbb{B}^{(m-1)}(\{t^\nu\}_2^{j-1}, \{\emptyset\}_j^{m-1})\right]_k = \delta_{jk} \left(\prod_{\nu=3}^{k-1} \frac{g^{(l)}(t^\nu, t^{\nu-1})}{f(t^\nu, t^{\nu-1})}\right) g^{(l)}(t^2, t^1). \tag{A.4}$$

Thus, taking into account that for $k > 1$, $T_{1,k}(u) = g^{(l)}(u, \xi) E_{k1}$ and

$$T_{1,k}(t^1) e_1 = g^{(l)}(t^1, \xi) e_k, \tag{A.5}$$

we immediately arrive at (A.1).

# B Comparison with known results of $U_q(\widehat{\mathfrak{gl}}_3)$ based models

Propositions 4.4 and 4.5 were already obtained for $m = 3$ in [46, 49], but using different normalization of Bethe vectors, and a different notation and normalization for the HC. We present here the connection between the two conventions. To clarify the presentation we will put a subscript *old* for the quantities dealt in [46, 49], and a subscript *new* for the ones used in the present article.

**Normalisation of (dual) Bethe vectors.** By comparison of their main terms, we get the following correspondence for Bethe vectors:

$$\mathbb{B}_{new}(\bar{t}) = \frac{\lambda_2(\bar{t}^2)}{\lambda_3(\bar{t}^2)} \mathbb{B}_{old}(\bar{t}^1, \bar{t}^2) \quad \text{and} \quad \mathbb{C}_{new}(\bar{s}) = \frac{\lambda_2(\bar{s}^2)}{\lambda_3(\bar{s}^2)} \mathbb{C}_{old}(\bar{s}^1, \bar{s}^2), \tag{B.1}$$

where $\bar{s} = \{\bar{s}^1, \bar{s}^2\}$ and $\bar{t} = \{\bar{t}^1, \bar{t}^2\}$. Note that in [46, 49], the sets $\bar{s}^1, \bar{s}^2$ and $\bar{t}^1, \bar{t}^2$ were noted $\bar{u}^C, \bar{v}^C$ and $\bar{u}^B, \bar{v}^B$ respectively.

---

[6]Obviously, $T_{i,j}(u) = g^{(l)}(u, \xi) E_{ji}$ for $i < j$.

**Sum formula.** Once the normalisation is fixed, one can compare the scalar product of Bethe vectors and the expressions given in proposition 4.4. In [49], the scalar product is expressed in term of functionals $r_1(z) = \alpha_1(z)$ and $r_3(z) = \alpha_2(z)^{-1}$. Using the normalisation (B.1), we get a sum formula identical to (4.8) with

$$W_{old}\begin{pmatrix} \bar{s}_{\text{I}}^1 & \bar{t}_{\text{I}}^1 & \bar{s}_{\text{II}}^1 & \bar{t}_{\text{II}}^1 \\ \bar{s}_{\text{I}}^2 & \bar{t}_{\text{I}}^2 & \bar{s}_{\text{II}}^2 & \bar{t}_{\text{II}}^2 \end{pmatrix} = f(\bar{s}^2, \bar{s}^1) f(\bar{t}^2, \bar{t}^1) W_{new}(\bar{s}_{\text{I}}, \bar{s}_{\text{II}} | \bar{t}_{\text{I}}, \bar{t}_{\text{II}}). \tag{B.2}$$

Note that in order to make the comparison, one has to exchange the subsets $\bar{s}_{\text{I}}^1 \longleftrightarrow \bar{s}_{\text{II}}^1$ in one of the sum formulas. This change is harmless since one performs a summation over all partitions $\bar{s}^1 \Rightarrow \{\bar{s}_{\text{I}}^1, \bar{s}_{\text{II}}^1\}$.

**Expression in term of HCs.** Applying the correspondence (B.2), the relation (4.11) is identical to the one obtained in [49] with

$$\begin{aligned} Z_{old}^{(l)}(\bar{s}^1, \bar{t}^1 | \bar{s}^2, \bar{t}^2) &= f(\bar{s}^2, \bar{s}^1) f(\bar{t}^2, \bar{t}^1) \, Z_{new}(\bar{s}^1, \bar{s}^2 | \bar{t}^1, \bar{t}^2), \\ Z_{old}^{(r)}(\bar{s}^1, \bar{t}^1 | \bar{s}^2, \bar{t}^2) &= f(\bar{s}^2, \bar{s}^1) f(\bar{t}^2, \bar{t}^1) \, \overline{Z}_{new}(\bar{s}^1, \bar{s}^2 | \bar{t}^1, \bar{t}^2). \end{aligned} \tag{B.3}$$

# C Coproduct formula for the dual Bethe vectors

The presentation (3.24) for the Bethe vector of the composite model can be treated as a co-product formula for the Bethe vector. Indeed, equation (3.20) formally determines a coproduct $\Delta$ of the monodromy matrix entries

$$\Delta(T_{i,j}(u)) = \sum_{k=1}^{m} T_{k,j}(u) \otimes T_{i,k}(u). \tag{C.1}$$

Then (3.24) is nothing but the action of $\Delta$ onto the Bethe vector.

The action of the coproduct onto the dual Bethe vectors can be obtained via antimorphism (3.16) thanks to the relation

$$\Delta_{q^{-1}} \circ \Psi = (\Psi \otimes \Psi) \circ \Delta'_q, \tag{C.2}$$

where

$$\Delta'_q(T_{i,j}(u)) = \sum T_{i,k}(u) \otimes T_{k,j}(u). \tag{C.3}$$

Then applying (C.2) to $\mathbb{B}_q(\bar{t})$, we get

$$\begin{aligned} \Delta_{q^{-1}}(\Psi(\mathbb{B}_q(\bar{t}))) &= \Delta_{q^{-1}}(\mathbb{C}_{q^{-1}}(\bar{t}^{-1})) = (\Psi \otimes \Psi) \circ \Delta'_q(\mathbb{B}_q(\bar{t})) \\ &= (\Psi \otimes \Psi)\left( \sum \frac{\prod_{\nu=1}^{N} \alpha_\nu^{(1)}(\bar{t}_{\text{I}}^\nu) f_q(\bar{t}_{\text{II}}^\nu, \bar{t}_{\text{I}}^\nu)}{\prod_{\nu=1}^{N-1} f_q(\bar{t}_{\text{II}}^{\nu+1}, \bar{t}_{\text{I}}^\nu)} \mathbb{B}_q(\bar{t}_{\text{I}}) \otimes \mathbb{B}_q(\bar{t}_{\text{II}}) \right) \\ &= \sum \frac{\prod_{\nu=1}^{N} \tilde{\alpha}_\nu^{(1)}(\frac{1}{\bar{t}_{\text{I}}^\nu}) f_q(\bar{t}_{\text{II}}^\nu, \bar{t}_{\text{I}}^\nu)}{\prod_{\nu=1}^{N-1} f_q(\bar{t}_{\text{II}}^{\nu+1}, \bar{t}_{\text{I}}^\nu)} \mathbb{C}_{q^{-1}}(\bar{t}_{\text{I}}^{-1}) \otimes \mathbb{C}_{q^{-1}}(\bar{t}_{\text{II}}^{-1}). \end{aligned} \tag{C.4}$$

Relabeling the subsets $\bar{t}_{\text{I}}^\nu \longleftrightarrow \frac{1}{\bar{t}_{\text{II}}^\nu}$ and using (5.25), we arrive at

$$\Delta_{q^{-1}}(\mathbb{C}_{q^{-1}}(\bar{t})) = \sum \frac{\prod_{\nu=1}^{N} \tilde{\alpha}_\nu^{(1)}(\bar{t}_{\text{II}}^\nu) f_{q^{-1}}(\bar{t}_{\text{I}}^\nu, \bar{t}_{\text{II}}^\nu)}{\prod_{\nu=1}^{N-1} f_{q^{-1}}(\bar{t}_{\text{I}}^{\nu+1}, \bar{t}_{\text{II}}^\nu)} \mathbb{C}_{q^{-1}}(\bar{t}_{\text{II}}) \otimes \mathbb{C}_{q^{-1}}(\bar{t}_{\text{I}}). \tag{C.5}$$

It remains to make the change $q^{-1} \to q$ to obtain (3.25). $\qquad\square$

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
