# Peer review of "Scalar products and norm of Bethe vectors for integrable models based on $U_q(\widehat{\mathfrak{gl}}_{n})$"

_SciPost Physics, doi:SciPost Phys. 4, 006 (2018)_

## Round 2 · Referee Report · Anonymous · 2017-12-21

Strengths

1. Necessary results for the form factor and correlation functions analysis of a wide range of models.
2. Encyclopaedic value. It is clear that any future paper on the algebraic Bethe ansatz approach to the higher rank algebras will be based on this paper.
2. Very useful and simple notations permitting to make extremely cumbersome computation readable

Weaknesses

1. Absence of the most important proof (the authors refer to their previous paper)

Report

It is an excellent paper giving first of all a systematic presentation of the algebraic Bethe ansatz approach to the higher rank integrable model with a trigonometric $R$ matrix (based on the $U_q(\widehat{\mathfrak{gl}_m})$). I'm quite sure that this paper will be a reference for most of the future work on such models. The proposition 4.4, the properties of the highest coefficient (Propositions 4.6-4.9) and the Gaudin formula (Theorem 4.10) are essential for any future analysis of the scalar products and form factors in the framework of the algebraic Bethe ansatz. I've specially appreciated the compact notations introduced by the authors permitting to write the extremely cumbersome results in a rather compact form.
The only weak point is the absence of the proof for the Gaudin formula. It is understandable that the authors don't want to increase the size of the paper repeating more or less directly the proof for the rational case, however a sketch of the proof (the main steps without technical details) would be helpful in my opinion.
In conclusion I think that the paper can be published in its present form in SciPost

Requested changes

I would like to suggest two optional changes:
1. in the beginning the authors introduce the off-shell Bethe vectors (3.6,3.7) without really giving a definition (referring to a previous paper). Afterwards they give however a nested construction for these vectors (5.8). It seems to me more logical to give this construction before the statement of the main results.
2. Add an appendix with the main steps of the proof of the Gaudin formula (probably without technical details)

---

## Round 2 · Referee Report · Anonymous · 2017-12-25

Strengths

1- The paper solves a longstanding problem.
2- Well presented.

Weaknesses

1- Part of a long sequence of papers and for this reason perhaps not easily accessible for non-experts.

Report

The paper solves a longstanding problem, namely to prove the generalization of the Gaudin-Korepin formula for the norm of Bethe wave functions to integrable models related to the quantum affine algebras $U_q (\widehat{\mathfrak{sl}_n})$. The authors also provide a combinatorial formula for the scalar product of off-shell Bethe vectors, generalizing old work of V. Korepin. In addition they derive recursion relations for off-shell Bethe vectors.

The paper is part of an effort to extend methods for the calculation of correlation functions of Yang-Baxter integrable models to higher-rank cases. It continues a long series of papers of some of the authors with various co-authors which was mostly dealing with rational R-matrices related to Yangian quantum groups. As in these rational cases it seems likely that the results presented in the paper will lead to the derivation of formulae for the matrix elements of local operators of various higher-rank integrable models parameterized in terms of Bethe roots. Such matrix elements appear, for instance, in the form-factor series of two-point correlation functions of local operators. In case of the XXZ spin chain such form-factor series could be analyzed numerically and asymptotically, which is the main motivation for trying a similar thing in the higher-rank case.

The paper is well presented and sufficiently self-contained. I recommend publication in the present form.

Requested changes

No changes necessary.

---

## Editorial Decision

published